# Pericyte-expressed Tie2 controls angiogenesis and vessel maturation

Martin Teichert[1,*,†], Laura Milde[1,*], Annegret Holm[1,†], Laura Stanicek[1,†], Nicolas Gengenbacher[1], Soniya Savant[1,2,†], Tina Ruckdeschel[1], Zulfiyya Hasanov[1,2], Kshitij Srivastava[1], Junhao Hu[1,†], Stella Hertel[1,2], Arne Bartol[1,2], Katharina Schlereth[1,2,**] & Hellmut G. Augustin[1,2,**]

The Tie receptors with their Angiopoietin ligands act as regulators of angiogenesis and vessel maturation. Tie2 exerts its functions through its supposed endothelial-specific expression. Yet, Tie2 is also expressed at lower levels by pericytes and it has not been unravelled through which mechanisms pericyte Angiopoietin/Tie signalling affects angiogenesis. Here we show that human and murine pericytes express functional Tie2 receptor. Silencing of Tie2 in pericytes results in a pro-migratory phenotype. Pericyte Tie2 controls sprouting angiogenesis in *in vitro* sprouting and *in vivo* spheroid assays. Tie2 downstream signalling in pericytes involves Calpain, Akt and FOXO3A. *Ng2-Cre*-driven deletion of pericyte-expressed Tie2 in mice transiently delays postnatal retinal angiogenesis. Yet, Tie2 deletion in pericytes results in a pronounced pro-angiogenic effect leading to enhanced tumour growth. Together, the data expand and revise the current concepts on vascular Angiopoietin/Tie signalling and propose a bidirectional, reciprocal EC-pericyte model of Tie2 signalling.

[1] Division of Vascular Oncology and Metastasis, German Cancer Research Center (DKFZ-ZMBH Alliance), Im Neuenheimer Feld 280, D-69120 Heidelberg, Germany. [2] Department of Vascular Biology and Tumor Angiogenesis (CBTM), Medical Faculty Mannheim, Heidelberg University, Ludolf-Krehl-Str. 13-17, D-68167 Mannheim, Germany. * These authors contributed equally to this work. ** These authors jointly supervised this work. † Present addresses: GE Healthcare, SE-751 84 Uppsala, Sweden (M.T.); Clinic for Paediatrics, University of Cologne, D-50924 Cologne, Germany (A.H.); Institute of Cardiovascular Regeneration, University of Frankfurt, D-60590 Frankfurt, Germany (L.S.); Department of Cell and Molecular Biology, Karolinska Institute, SE-171 77 Stockholm, Sweden (S.S.); Interdisciplinary Research Center on Biology and Chemistry, Chinese Academy of Sciences, CN-200031 Shanghai, China (J.H.). Correspondence and requests for materials should be addressed to H.G.A. (email: augustin@angiogenese.de).

The formation of new blood vessels from existing vessels, known as angiogenesis, depends on a sequential process that involves endothelial cells (EC) and pericytes[1]. The VEGF-dependent initiation of endothelial sprouting requires the detachment of pericytes to enable EC migration. Vice versa, pericyte recruitment by nascent vessels occurs during the later steps of the angiogenic cascade providing stabilization and maturation signals[2]. Pericytes contribute to vessel maturation by direct, basement-membrane penetrating contacts with EC and through the release of paracrine-acting growth factors. The extent of pericyte coverage is an important marker of vessel maturation[3]. Yet, pericyte coverage in resting vessels varies widely in different organs and with type of microvasculature (continuous versus discontinuous versus fenestrated). Different molecules have been proposed as pericyte markers, including PDGFRb, CSPG4 (NG2), ACTA2 (SMA), Desmin and CD248 (refs 3–6). The expression of these markers depends on tissue, type of vessel and maturation state.

PDGFb, secreted by angiogenic EC, acts as the best characterized growth factor recruiting PDGFRb expressing pericytes[7]. Angiopoietin/Tie (Ang/Tie) signalling similarly controls the association of EC and pericytes[8]. Yet, its molecular mechanism-of-action is much less well understood. Tie2

expressed by EC is activated by pericyte-expressed Ang1, which contributes to maintaining the quiescent EC phenotype. Ang1-driven phosphorylation of Tie2 results in the activation of downstream pathways mediating survival, proliferation, migration and anti-inflammatory signals. Importantly, vessel stabilization/survival involves AKT-dependent FOXO1 regulation[9]. In contrast, the second angiopoietin ligand, Ang2 acts as partial agonist of Tie2, that is, it inhibits Tie2 signalling in the presence of Ang1 and weakly activates Tie2 in the absence of Ang1 (ref. 10). Notably, Ang2 is produced, stored and upon stimulation released by EC, thereby acting as autocrine regulator of Ang/Tie2 signalling[11]. Recombinant or transgenic Ang2 application results in pericyte drop-out in the mouse retina leading to vessel destabilization[12].

Taken together, there is strong evidence that Ang/Tie2 signalling affects pericyte function, but the established endothelial-centric model of paracrine acting Ang1 and autocrine acting Ang2 is not sufficient to explain how Ang/Tie2 signalling affects pericyte function and EC-pericyte association. EC may be the primary target cell of Ang/Tie2 signalling. However, work performed during the last 10 years has provided evidence that Tie2 is not exclusively an EC-specific receptor. Tie2 expression and function has been established in hematopoietic stem cells[13],

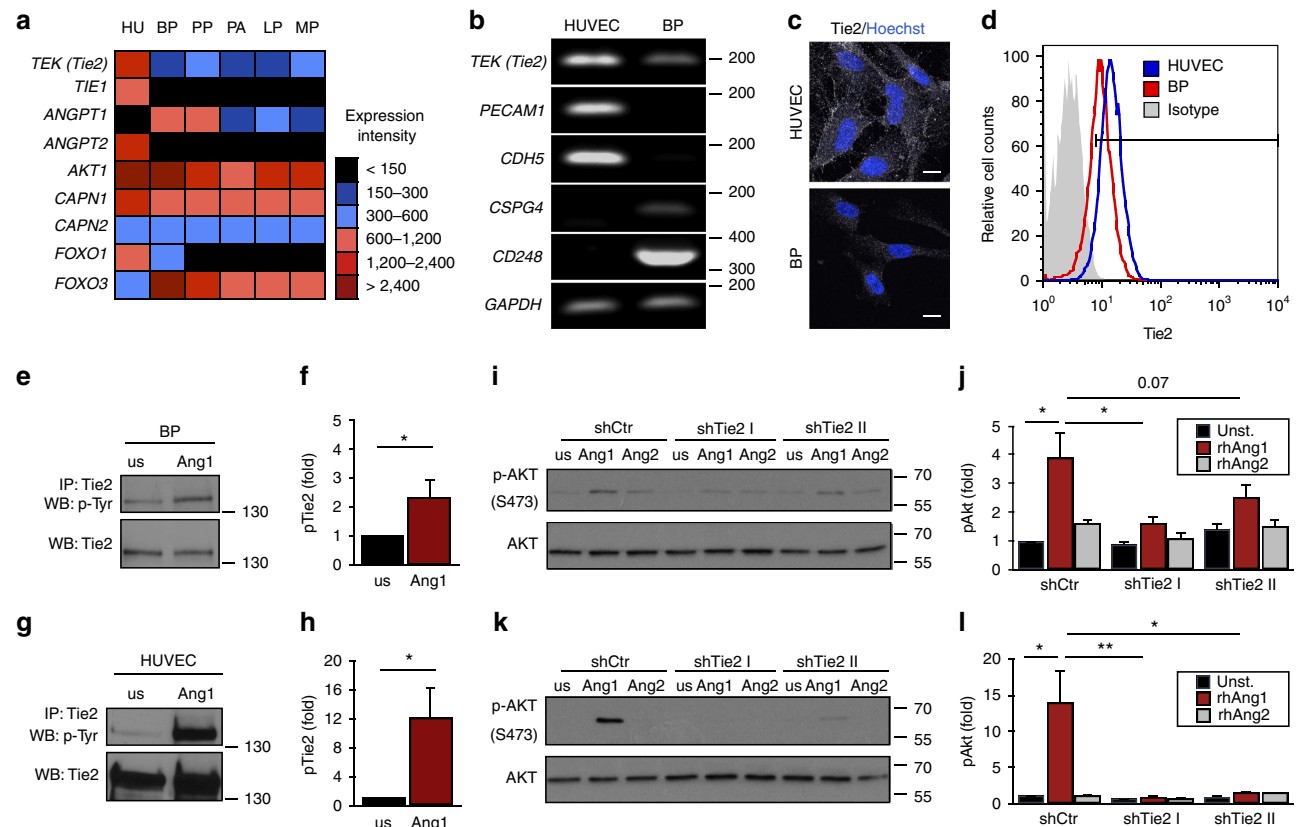

**Figure 1 | Pericytes express functional Tie2. (a)** Microarray-based expression screening of brain pericytes (BP), placenta pericytes (PP), pancreas pericytes (PA), lung pericytes (LP), muscle pericytes (MP) and human umbilical vein endothelial cells (HUVEC [HU]). **(b)** Semi-quantitative PCR of *TEK (Tie2)*, endothelial marker genes (*PECAM1, CDH5*) and pericyte marker genes (*CSPG4, CD248*) in HUVEC and BP; house keeping gene: *GAPDH*. **(c)** Representative images showing expression of Tie2 in HUVEC and BP. **(d)** Histogram of membrane-bound Tie2 expression in HUVEC and BP compared to isotype control measured by flow cytometry. **(e,g)** Western blot (WB) analysis of tyrosine phosphorylation (pTyr) and total Tie2 after Tie2 immunoprecipitation (IP) in BP **(e)** and HUVEC **(g)** upon stimulation with recombinant human (rh) Ang1 compared to unstimulated (us) cells. **(f,h)** Quantification of the ratio of phosphorylated Tie2 (pTie2) relative to total Tie2 protein in BP **(f)** and HUVEC **(h)** normalized to us control (n = 3). **(i,k)** Western blot analysis of AKT phosphorylation (Ser473) in control (shCtr) and Tie2-silenced (shTie2 I/shTie2 II) BP **(i)** and HUVEC **(k)** upon stimulation with rhAng1 or rhAng2. **(j,l)** Quantification of the ratio of pAKT to total AKT protein expression in BP **(j)** and HUVEC **(l)** normalized to unstimulated control (n = 3). Representative western blot images are cropped versions and original images can be found in Supplementary Fig. 15. Scale bars: 20 μm **(c)**. Data are shown as mean ± s.d. Statistics were performed using Mann–Whitney U test (f,h) and one-way ANOVA (j,l). *P < 0.05, **P < 0.01.

macrophages[14], muscle satellite cells[15], neural cells[16] and tumour cells[17,18]. Notably, scattered reports have described Tie2 expression by vascular smooth muscle cells and pericytes[19,20].

We have in the present study validated weak, but consistently detectable expression of functional Tie2 receptor on different pericyte populations. Based on these findings, we hypothesized that pericyte-expressed Tie2 may directly control pericyte function. Cellular, biochemical and genetic experiments collectively establish a role of pericyte-expressed Tie2 in the control of angiogenesis and vascular maturation. Genetic inactivation of Tie2 in pericytes leads to strongly enhanced tumour growth, demonstrating that Tie2 in pericytes affects the plasticity window of the tumour-associated vasculature.

## Results

**Pericytes express functional Tie2 receptor.** To assess the expression of Tie2 and other Ang/Tie signalling molecules on human pericytes, a microarray-based expression profiling of human brain pericytes (BP), placenta pericytes, pancreas pericytes (PA), lung pericytes (LP), muscle pericytes (MP) and human umbilical vein endothelial cells (HUVEC) was performed. *TEK* (*Tie2*) expression was highest in HUVEC, but also detectable at lower level in all analysed pericyte lines (Fig. 1a). The microarray data were also used to validate the identity of the employed pericyte populations (Supplementary Fig. 1). We included human dermal fibroblasts (Fib) in these validation experiments in recognition of the overlapping marker profiles of pericytes and fibroblasts. Indeed, Fib cells expressed with some quantitative variations the same markers, but no detectable *TEK* (*Tie2*) (Supplementary Fig. 1).

For further functional experiments, BP were selected as prototypic pericyte population and expression of the different marker genes was validated by semi-quantitative PCR (Fig. 1b). BP expressed *TEK* (*Tie2*) as well as the established pericyte markers *CSPG4/NG2* and *CD248*, but not the EC markers *PECAM1* and *CDH5*. Immunocytochemistry and FACS analysis identified lower, albeit consistently detectable levels of Tie2 protein by BP compared to EC (Fig. 1c,d). Stimulation of BP with the Tie2 ligand Ang1 resulted in detectable Tie2 phosphorylation, albeit again at considerably lower levels as in EC (Fig. 1e–h). Ang1, but not Ang2, induced phosphorylation of the Tie2 downstream target AKT in BP (Fig. 1i,j) as well as in EC (Fig. 1k,l). AKT phosphorylation was Tie2-dependent as evidenced by shRNA-mediated Tie2 silencing experiments (Fig. 1i–l, Supplementary Fig. 2a,b). Lentiviral-based transduction had no influence on the expression of pericyte markers (Supplementary Fig. 2c,d). Collectively, the data show that human pericytes express low level of functional Tie2 receptor.

**Pericyte Tie2 loss is phenocopied by Ang2 gain.** We next set out experiments aimed at assessing direct effects of Tie2 signalling on pericyte function and signal transduction. Consistent with the primary role of Tie2 in survival signalling and quiescence of EC, Ang1 stimulation of BP was robustly anti-apoptotic, but had no effect on pericyte proliferation or migration (Supplementary Fig. 3). Yet, pericytes themselves have been shown to be a primary source of Ang1 (refs 21–23) (which was confirmed in the microarray analysis (Fig. 1a)), suggesting an autocrine mode of action. We therefore performed lateral scratch wound assays with Tie2 expressing compared to Tie2 shRNA-silenced pericytes. Silencing of Tie2 expression significantly increased pericyte motility (Fig. 2a,b). Similar results were obtained with siRNA (Supplementary Fig. 4a,b).

Ang2 is a partial agonist of Tie2 that acts as functional antagonist of Ang1 in Tie2 expressing cells[8,10]. Moreover, Ang2

expression is strongly upregulated in EC during sprouting angiogenesis[24,25]. We consequently hypothesized that EC-derived Ang2 could affect pericyte migration by functionally antagonizing Tie2 signalling. Stimulation of pericytes with supernatants from either Ang2 overexpressing HUVEC or supernatants from control HUVEC resulted in significantly increased pericyte migration following Ang2 stimulation (Fig. 2c, Supplementary Fig. 4c,d). Taken together, the Tie2 pericyte loss-of-function phenotype is phenocopied by the Ang2 gain-of-function phenotype.

To mechanistically assess Tie2 function in pericytes, we biochemically analysed migration-associated molecules. Cultured pericytes abundantly expressed calpain1 (Fig. 2d) and showed a significant induction of calpain1 activity after stimulation with supernatants of Ang2 overexpressing HUVEC (Fig. 2e). FOXO transcription factors are important downstream molecules of Tie2-mediated AKT signal transduction[26,27]. FOXO3A has been shown to be regulated by calpain1 (ref. 28). Our microarray experiments had identified strong *FOXO3A* and weak *FOXO1* expression in pericytes (Fig. 1a). Western blot experiments verified strong FOXO3A expression in pericytes, but only weak expression in EC (Supplementary Fig. 4e). We consequently stimulated cultured BP with either Ang1 or Ang2, to study if Tie2 signalling in pericytes controlled FOXO3A phosphorylation. Ang1 stimulation resulted in increased FOXO3A phosphorylation, whereas Ang2 reduced FOXO3A phosphorylation (Fig. 2f). Phosphorylation leads to FOXO3A inactivation and cytoplasmic retention[29]. Correspondingly, immunocytochemical analysis of FOXO3A localization identified prominent nuclear FOXO3A in cultured pericytes upon Ang2 stimulation (Fig. 2g). The data suggest that Tie2 in pericytes influences FOXO3A-dependent signalling.

To further assess the effect of Tie2 in pericytes on EC-pericyte intercellular signalling, we performed a phospho-receptor tyrosine kinase array of BP and HUVEC co-cultures. Eph receptor and ephrin ligand signalling has been implicated in vascular development and in *in vivo* models of angiogenesis where bi-directional signalling mediates juxtracrine cell–cell contact, cell adhesion to extracellular matrix and cell migration[30–32]. Furthermore, Eph signalling in mural cells has been shown to control cell motility and adhesion as well as pericyte-EC assembly[33,34]. Both EC and pericytes expressed Eph receptors and ephrin ligands (Supplementary Fig. 5). Co-culture of Tie2-silenced pericytes with EC resulted in reduced EphA2, EphB2 and EphB4 phosphorylation in the co-culture lysates (Fig. 2h). Taken together, the cellular experiments identified a Tie2-dependent control of key regulators of cellular migration and invasion (calpain1, Eph and FOXO3A) in pericytes.

**Tie2-expressing pericytes limit endothelial sprouting.** We next assessed, if Tie2-silenced pericytes affected the sprouting of EC in spheroid sprouting angiogenesis assays. In this assay, co-culture spheroids of EC with WT or Tie2-silenced pericytes were embedded in a gel matrix and EC sprouting was quantitated after 24 h (Fig. 3a). Tie2 expressing pericytes significantly inhibited EC sprouting (Fig. 3b). To translate these findings into an *in vivo* setting, we performed a modification of the established *in vivo* spheroid-based angiogenesis assay[35], by co-grafting HUVEC with WT or Tie2-silenced BP in a gel matrix subcutaneously into immunocompromised mice. Analysis of vascularization after 21 days identified a significant reduction of EC microvessel density (MVD) (CD34 expression) in plugs grafted with Tie2-expressing pericytes compared to Tie2-silenced pericytes (Fig. 3c,d). Together, these findings demonstrate that Tie2-expressing pericytes limit in a paracrine manner EC sprouting angiogenesis.

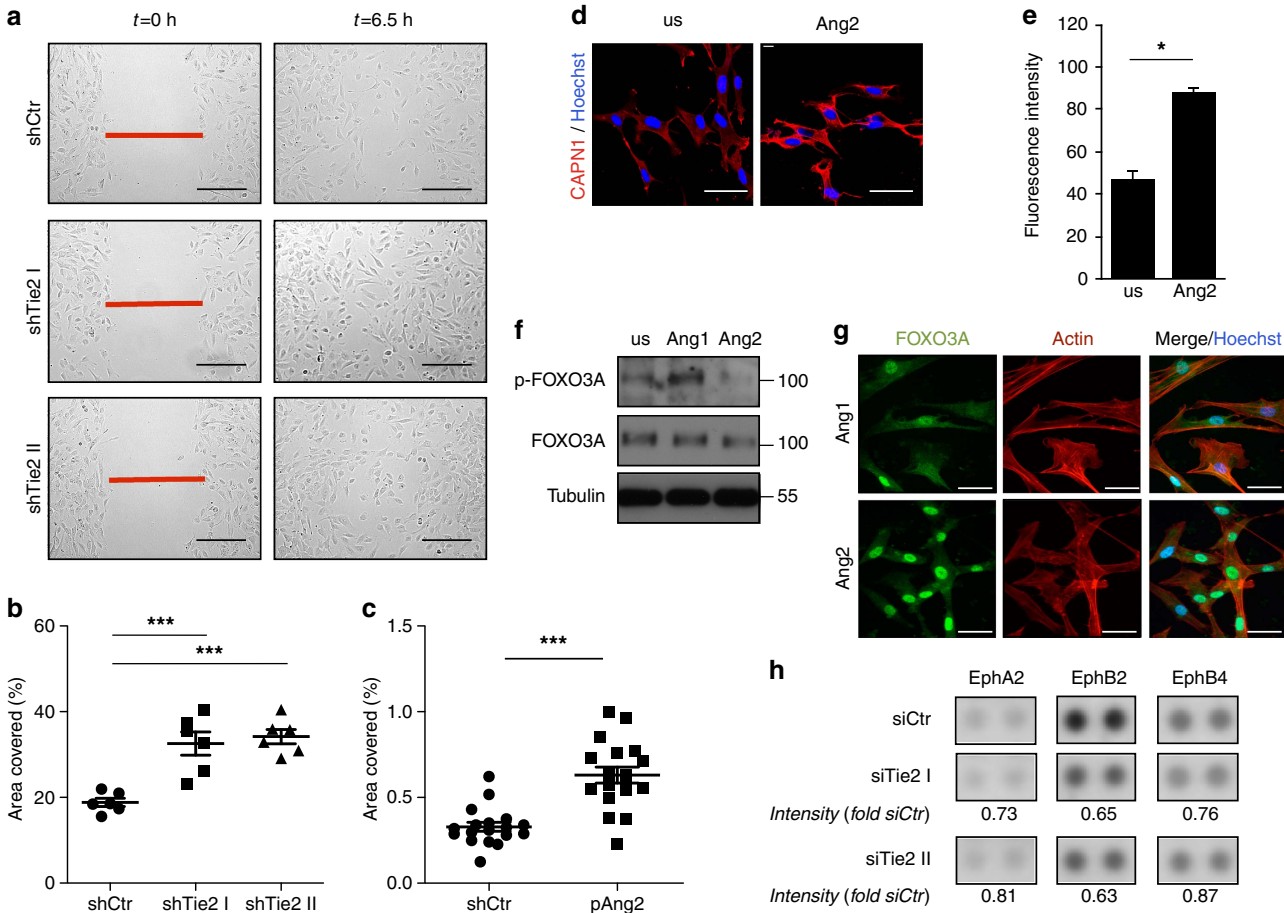

**Figure 2 | Tie2-dependent signalling controls pericyte migration.** (**a**) Representative images of control BP (shCtr) and silenced for Tie2 (shTie2 I and shTie2 II) at the beginning ($t = 0$ h) and end ($t = 6.5$ h) of time-lapse microscopy during lateral-scratch wound assay. (**b**) Quantification of covered area by BP shTie2 I and shTie2 II in a lateral-scratch wound assay after 6.5 h normalized to BP shCtr. (**c**) Quantification of covered area by BP treated with supernatants of control (Ctr) or Ang2-overexpressing (pAng2) HUVEC in a lateral scratch wound assay after 6.5 h normalized to control. (**d**) Representative images of calpain 1 (CAPN1) staining of BP stimulated with recombinant human (rh) Ang2 for 1 h or left unstimulated (us). Nuclei were counterstained with Hoechst. (**e**) Quantification of calpain 1 activity measured by detection of fluorescent cleaved substrate in BP stimulated for 1 h with rhAng2 normalized to unstimulated (us) control ($n = 3$). (**f**) Western blot analysis of phosphorylated FOXO3A (pFOXO3A) and total FOXO3A in BP stimulated with rhAng1 or rhAng2 for 30 min. Tubulin served as loading control. (**g**) Representative images showing FOXO3A localization in BP stimulated with rhAng1 or rhAng2 for 30 min. Cells were further stained with actin (cytoskeleton staining) and Hoechst (nucleus staining). (**h**) Human phospho-receptor tyrosine kinase array for EphA2, EphB2 and EphB4 ephrin receptors from control BP (shCtr) and BP silenced for Tie2 (siTie2 I and siTie2 II). Intensity is shown as fold of siCtr. Representative western blot and array images are cropped versions and original images can be found in Supplementary Figs 15 and 17. Scale bars: 250 μm (**a**), 50 μm (**d,g**). Data are shown as mean ± s.d. Statistics were performed using one-way ANOVA (**b**), Mann–Whitney $U$ test (**c**) and Student's $t$-test (**e**). *$P < 0.05$, ***$P < 0.001$.

**Tie2 deletion in pericytes delays postnatal angiogenesis**. To validate pericyte Tie2 expression *in vivo*, we crossed *Ng2-Cre* mice with YFP reporter mice[36] and isolated YFP-positive NG2-expressing cells from the lungs (Supplementary Fig. 6a). Relative mRNA quantification via qPCR revealed that NG2-positive cells were devoid of *Pecam1* expression, but harboured *Tek* (*Tie2*) expressing cells (Fig. 4a). To trace the identity of the *Ng2-Cre* labelled cells, we performed whole mount immunofluorescence analyses of postnatal retinas stained with the EC marker isolectinB4. NG2-positive cells ensheathed the isolectinB4 positive endothelium with pronounced coverage in arterioles, capillaries and venules (Supplementary Fig. 6b). Together, these reporter gene experiments confirmed the expression of Tie2 by pericytes and further validated *Ng2-Cre* as a robust pan-pericyte driver for *in vivo* mutagenesis studies.

In order to functionally study the role of Tie2 in pericytes *in vivo*, mice with a conditional Tie2 allele (*Tie2*^fl/fl)[37] were crossed to *Ng2-Cre* mice (*Tie2*^PEKO) to generate mice with constitutively deleted Tie2 in pericytes. Pericyte-specific Tie2 deletion was validated by marker gene analysis of EC and pericytes isolated from the lungs of *Ng2-Cre* x YFP x *Tie2*^fl/fl mice (Fig. 4b). While *Tek* (*Tie2*) expression in EC of these mice was not affected, pericyte *Tek* (*Tie2*) mRNA expression was reduced by 50–60%. *Pdgfrb* and *Pecam1* served as pericyte and EC marker genes, respectively, whose expression was not altered (Fig. 4b). Western blot and FACS analyses of isolated BP from WT and *Tie2*^PEKO mice confirmed reduced Tie2 protein expression in pericytes (Supplementary Fig. 7).

*Tie2*^PEKO mice were born close to the predicted Mendelian ratio and developed normally (Supplementary Fig. 8). Whole mount retinal image analysis of P4, P6 and adult mice identified a transient delay of retinal vascularization characterized by a reduction of MVD at peak retinal angiogenesis (P4; at this stage no difference in pericyte coverage) and a reduction of pericyte coverage upon completion of lateral angiogenesis (P6; at this stage no difference in MVD) (Fig. 4c–f). There was no detectable

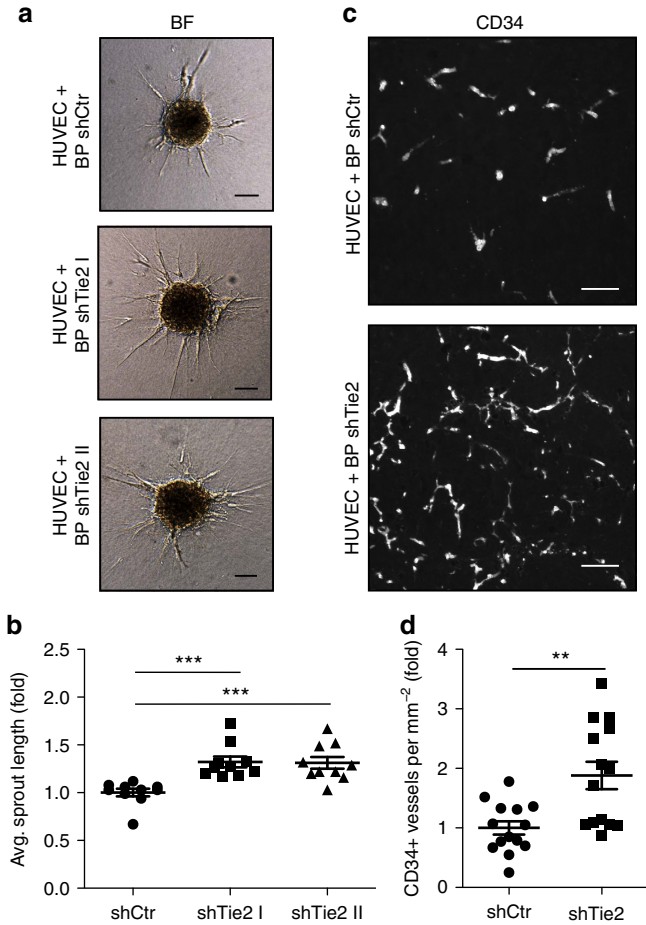

**Figure 3 | Tie2-expressing pericytes limit endothelial sprouting.**
(**a**) Representative bright field (BF) images of *in vitro* sprouting assay using HUVEC and BP + shCtr or HUVEC and BP + shTie2 co-culture spheroids, respectively. (**b**) Quantification of average sprout length in *in vitro* sprouting assay using HUVEC-BP shTie2 co-culture spheroids normalized to HUVEC-BP shCtr co-culture spheroids. (**c**) Representative images of *in vivo* plug assay sections using HUVEC-BP shCtr and HUVEC-BP shTie2 co-culture spheroids stained for CD34. (**d**) Quantification of microvessel density (MVD) in CD34-stained plugs using HUVEC-BP shTie2 co-culture spheroids normalized to HUVEC-BP shCtr co-culture spheroids. Scale bars: 100 μm (**a**), 50 μm (**c**). Data are shown as mean ± s.d. Statistics were performed using one-way ANOVA (**b**) and Mann–Whitney *U* test (**d**). **$P < 0.01$, ***$P < 0.001$.

quantitative and qualitative difference in retinas of adult mice (Fig. 4c–f). Taken together, pericyte-specific deletion of Tie2 results in a transient of angiogenesis and vessel maturation during physiological postnatal angiogenesis.

**Tie2 deletion in pericytes enhances tumour growth.** We next asked if the rather mild postnatal phenotype in *Tie2^PEKO^* mice affected the vascularization of tumours and possibly tumour growth. B16 melanoma cells or Lewis lung carcinoma (LLC) cells were subcutaneously grown in WT and *Tie2^PEKO^* mice. In both models, tumour growth was strongly enhanced in the absence of Tie2 on pericytes (Fig. 5a–d). In order to determine how Tie2-deficient pericytes affected tumour growth kinetics, Luciferase-expressing B16 melanomas were grown s.c. in WT and *Tie2^PEKO^* mice and tumour growth was traced non-invasively over time (Fig. 5e). The experiment confirmed the enhanced tumour growth in *Tie2^PEKO^* mice (Fig. 5f). Yet, when plotting log-

transformed tumour growth data to assess tumour growth rates, these curves diverged at early stages of tumour growth. At later time points, tumour growth curves were parallel indicating identical tumour growth rates (Fig. 5g). Together, the tumour experiments identified a profound gain of function phenotype in mice with targeted deletion of Tie2 in pericytes.

**Tie2 deletion in pericytes enhances tumour angiogenesis.** Morphological analysis of the tumours revealed significant differences in tumour vascularization in *Tie2^PEKO^* mice: MVD was significantly higher in B16 as well as LLC tumours (Fig. 6a,b). The average vessel diameter was smaller in KO tumours (Fig. 6c). Co-localization analysis of pericytes (NG2) and EC (CD31) identified a moderate, albeit non-significant reduction of pericyte coverage in tumours grown in *Tie2^PEKO^* mice (Fig. 6d,e). Similar data were obtained when using Desmin as pericyte marker (Supplementary Fig. 9). As seen in *in vitro* co-culture studies, EphA2, EphB2 and EphB4 phosphorylation was reduced in whole tumour lysates of *Tie2^PEKO^* animals, independently of changes in gene expression (Supplementary Fig. 10). In turn, expression of VEGF or the VEGF receptors and their phosphorylation was not altered (Supplementary Fig. 11).

To assess how the vascular changes in *Tie2^PEKO^* mice would affect the perfusion and permeability of tumour vessels, we analysed tissue distribution of tail vein-injected rhodamine-labelled fluorescent latex microspheres. Significantly more microspheres accumulated in tumours grown in *Tie2^PEKO^* mice 24 h after sphere injection (Fig. 6f–h). Notably, transmission electron microscopy confirmed perfusion by detecting erythrocytes in the smaller diameter *Tie2^PEKO^* vessels (Supplementary Fig. 12a). Perfusion of microvessels in *Tie2^PEKO^* mice was further confirmed by similar extent of hypoxic areas as compared to tumours growth in wild-type mice (Supplementary Fig. 12b). Tracing of lung metastasis following surgical removal of the primary tumour identified more detectable metastases in *Tie2^PEKO^* mice compared to WT mice (Supplementary Fig. 12c,d). Collectively, the experiments identified a more angiogenic and leaky, yet well perfused vasculature in tumours of mice with targeted deletion of Tie2 in pericytes.

**Discussion**
The Angiopoietin-Tie system was originally discovered as EC-specific signalling pathway, playing a fundamental role during embryonic vessel assembly and maturation as well as vessel maintenance in adults[8]. Indeed, global deletion of Tie2 results in embryonic lethality around E10.5 as a result of remodelling and maturation defects of the embryonic vasculature[38]. Correspondingly, endothelial-specific deletion of Tie2 using *Ve-cad-Cre* driver mice, pursued in validating the Tie2^fl/fl^ mice used in this study, essentially phenocopied the global Tie2 KO phenotype with embryonic lethality around E10.5 (Supplementary Fig. 13), confirming the key role of endothelial Tie2 during embryonic development. Tie2 is, however, not exclusively expressed by EC[13–20]. Based on the results of a systematic expression profiling screen of cultured pericytes isolated from different organs (Fig. 1a), the present study was aimed at analysing the contribution of pericyte-expressed Tie2 to vascular function. Employing a combination of cellular, biochemical and genetic experiments, we showed that (i) human and murine pericytes express functional Tie2 receptor, (ii) Tie2-silenced pericytes have a pro-migratory phenotype, (iii) Tie2 downstream signalling in pericytes involves Calpain, Akt and FOXO3A, (iv) *Ng2-Cre*-driven deletion of pericyte-expressed Tie2 delays developmental angiogenesis and vessel maturation, and (v) Tie2 deletion in

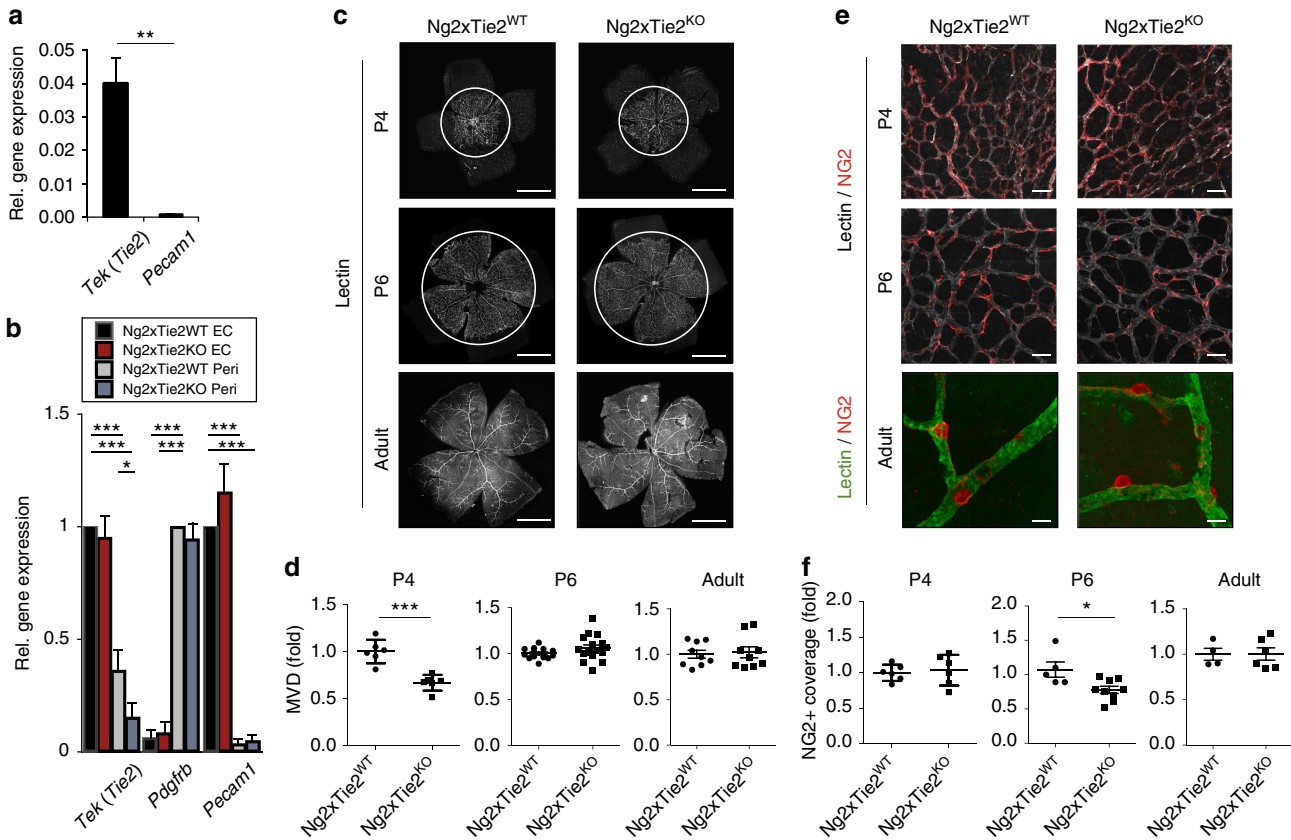

**Figure 4 | Tie2 deletion in pericytes delays postnatal angiogenesis and vessel maturation.** (**a**) Quantitative PCR (qPCR) demonstrating *Tie2* and *Pecam1* expression in sorted YFP-positive ( = NG2-positve) cells isolated from the lungs of *Ng2-Cre* x YFP mice. Expression is normalized to *B2m* mRNA expression (*n* = 3). (**b**) Analysis of *Tie2, Pdgfrb* and *Pecam1* expression by qPCR in EC and pericytes isolated from *Tie2*[PEKO] or WT mice, respectively (*n* = 3). (**c**) Representative images of the retinal vasculature stained with isolectin B4 at postnatal day 4 (P4), P6 and in adult WT and *Tie2*[PEKO] mice. (**d**) Quantification of retinal microvessel density (MVD) in P4, P6 and adult retinas of *Tie2*[PEKO] mice normalized to WT littermate average. (**e**) Representative images of retina vessels stained for NG2 and with isolectin B4 in P4, P6 and adult WT and *Tie2*[PEKO] mice. (**f**) Quantitative analysis of NG2 coverage of isolectin-positive retinal vessels in P4, P6 and adult *Tie2*[PEKO] mice normalized to WT littermate average. Scale bars: 1 mm (**c**), 50 μm (**e** (P4, P6)), 20 μm (**e** (adult)). Data are shown as mean ± s.d. Statistics were performed using Student's *t*-test (**a**), one-way ANOVA (**b**) and Mann–Whitney *U* test (**d,f**). *$P < 0.05$, ** $P < 0.01$, ***$P < 0.001$.

pericytes results in a pro-angiogenic tumour vasculature with enhanced tumour growth. Together, the data expand and revise the current concepts on vascular Ang-Tie signalling and propose a bidirectional, reciprocal EC-pericyte model of Tie2 signalling (Fig. 7).

The results of the present study have shed unexpected light into the mechanisms of vascular Ang/Tie signalling that may help to answer a number of enigmatic questions in the Ang/Tie field and that may have implications for the therapeutic exploitation of Ang-Tie signalling, particularly in the context of tumour angiogenesis. First, we identified Tie2 expression in cultured pericytes isolated from different organs and established cellular and signalling readouts of Ang-Tie signalling in cultured pericytes. These findings hinted at a functional role of pericyte-expressed Tie2 in regulating vascular function. However, the intensity of Tie2 expression and pTie2 level were considerably lower compared to EC. Therefore, only the conditional cell type-specific genetic *in vivo* approach yielded definite and unambiguous insights into pericyte Tie2 function. We employed towards this end *Ng2-Cre* mice[39] as most robust and cell type selective driver for pericyte-specific mutagenesis experiments. The choice of *Ng2-Cre* was based on reporter gene experiments (Supplementary Fig. 6) as well as the surprising finding that *Pdgfrb-Cre* mice[33] did not prove suitable to pericyte-specifically delete Tie2. In fact, *Pdgfrb-Tie2*[KO] mice died around E10.5

similar to *Ve-cad-Tie2*[KO]. Analysis of *Pdgfrb-Cre*-mediated conditional mutagenesis revealed germline expression of *Pdgfrb-Cre* in both male and female mice resulting first in a heterozygous and following a second round of breeding a homozygous global deletion of *Tie2* (Supplementary Fig. 14). Thus, efforts to breed *Pdgfrb-Tie2*[KO] mice homozygous on the *Tie2* allele resulted in death of homozygous mice due to a global Tie2 deletion at E10.5 similar as it has been described before[38]. *Ng2-Cre* is known to be expressed in the male germline, which needs to be considered in using *Ng2-Cre* mice for cell type-specific conditional mutagenesis experiments, but the germline expression of *Pdgfrb-Cre* in both sexes limits the use of *Pdgfrb-Cre* as deleter for pericyte-related experiments.

Second, Tie2 deletion in pericytes caused a mild and transient developmental delay of postnatal retinal angiogenesis and vessel maturation. In turn, tumour experiments in pericyte Tie2 null mice resulted in a sustained pro-angiogenic tumour vessel phenotype with strongly enhanced tumour growth. This apparent difference between developmental and tumour angiogenesis may be related to differences in physiological and pathological angiogenesis. Embryonic and postnatal angiogenesis is a developmental program leading to a remodelled and fully mature vasculature. In this scenario, pericyte Tie2 deletion leads to a transient delay first, of angiogenesis (P4) and second, of vessel maturation (P6). Yet, this delay of retinal angiogenesis can be

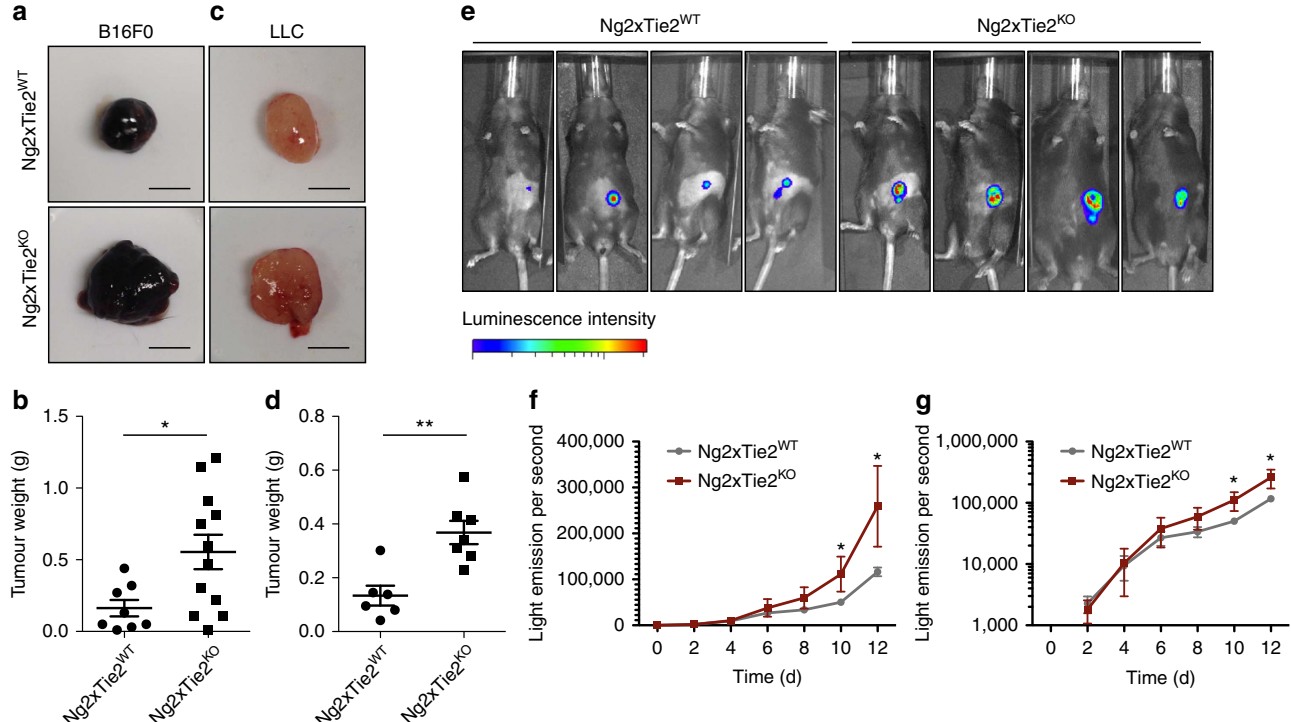

**Figure 5 | Pericyte-specific deletion of Tie2 enhances tumour growth.** (**a**) Representative images of B16 tumours grown subcutaneously for 14 days in WT and *Tie2^PEKO* mice. (**b**) Quantification of tumour weight of B16 tumours grown subcutaneously for 14 days in WT and *Tie2^PEKO* mice. (**c**) Representative images of LLC tumours grown subcutaneously for 12 days in WT and *Tie2^PEKO* mice. (**d**) Quantification of tumour weight of LLC tumours grown subcutaneously for 12 days in WT and *Tie2^PEKO* mice. (**e**) Representative images from bioluminescence analysis 10 days after subcutaneous injection of B16 cells in WT and *Tie2^PEKO* mice. (**f**) Quantitative bioluminescent analysis (linear) of B16 tumour burden in WT and *Tie2^PEKO* mice. (**g**) Quantitative bioluminescent analysis (log-transformed) of B16 tumour burden in WT and *Tie2^PEKO* mice. Scale bars: 5 mm (**a**,**c**). Data are shown as mean ± s.d. Statistics were performed using Student's *t*-test. *$P < 0.05$, **$P < 0.01$.

compensated and pericyte Tie2 appears to be dispensable for homeostatic maintenance of the vasculature in adult mice. In turn, tumour angiogenesis is characterized by a pathological patency of the angiogenic stimulus resulting in a chaotropic, pro-angiogenic and permanently immature vasculature. As a result, loss of pericyte Tie2 had a durable effect that was not compensated over time. The tumour phenotype thereby corresponded to the cellular and spheroid co-culture transplantation experiments that had revealed a restraining phenotype of Tie2 expressing pericytes on sprouting angiogenesis (Figs 2 and 3). Notably, the higher density of smaller diameter blood vessels in the tumour experiments resulted in enhanced tumour growth. This is in contrast to the complete ablation of pericytes in tumours. The Ng2 promoter-driven thymidine kinase-mediated (tk) depletion of pericytes has recently been shown to result in reduced tumour angiogenesis and correspondingly tumour growth[40,41]. The results of this study suggest that loss of pericyte Tie2 distinctly interferes with EC–pericyte interactions and does not globally impair all pericyte-mediated vascular stabilization functions.

Third, the tumour phenotype of mice with genetic deletion of Tie2 in pericytes warrants some careful reconsideration of the concepts of therapeutic angiopoietin targeting to interfere with tumour angiogenesis and for the future design of anti-angiogenic combination therapies. The genetic strategy employed in the present study had yielded on average a reduction of pericyte Tie2 mRNA and protein of only around 50%. Yet, this deletion was sufficient to yield a substantial acceleration of tumour growth. If anything, the incomplete recombination of Tie2 in pericytes would have underestimated the effect of Tie2 deletion on tumour

angiogenesis. Ang/Tie pathway targeting therapies have so far focussed on Ang2 whose inhibition affects tumour angiogenesis by contextually interfering with EC sprouting angiogenesis and by promoting vessel maturation (and thereby normalization) of the remodelling, pericyte-associating vasculature[42]. This model may still be correct, but the therapeutic mechanism may not solely be due to EC targeting effects. Intriguingly, the pericyte Tie2 KO tumour growth and tumour vascular phenotype has striking parallels to the phenotype of tumours grown in Ang2-deficient mice[43]. Notably, the loss of Ang2 results in a transient reduction of tumour growth rates[43]. The tumour phenotype in the present study was correspondingly characterized by a transient increase in tumour growth rates. The apparent temporally identical and phenotypically opposing tumour vessel phenotypes may relate to the cellular findings of the present study, in which the pericyte Tie2 loss-of-function phenotype phenocopied the Ang2 gain of function. The findings of the present study make it likely that the benefit of pharmaceutical Ang2 targeting may not just affect EC, but at least in part also be attributed to targeting pericyte Ang/Tie signalling.

Fourth, the cellular and biochemical analyses of the present study suggest that paracrine Ang2 (produced by EC) may in an antagonistic manner interfere with autocrine Ang1/Tie2 signalling in pericytes. Ang2 has been shown to act as partial agonist of Tie2, that is, it is an antagonist of Tie2 activation in the presence of Ang1 and a weak agonist in the absence of Ang1 (ref. 10). Moreover, the orphan receptor Tie1 has been suggested to be involved in Ang2 agonistic effects on Tie2 (ref. 44). This concept has recently been validated in definite *in vivo* experiments establishing a critical role of Tie1 in mediating agonistic effects of

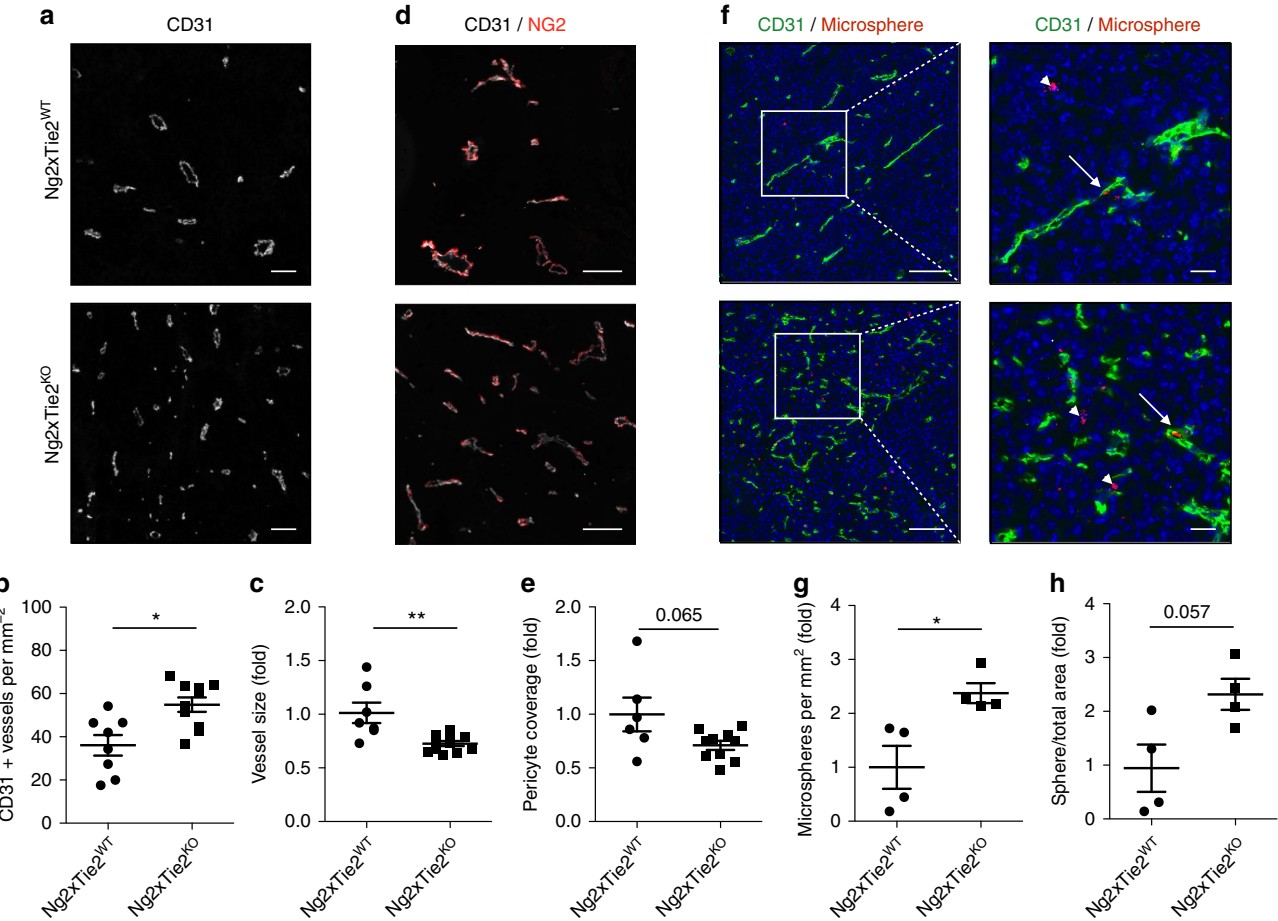

**Figure 6 | Tie2 deletion in pericytes enhances tumour angiogenesis.** (**a**) Representative images of B16 tumour vasculature stained with CD31 in WT and *Tie2^PEKO* mice. (**b**) Quantification of microvessel density (MVD) in CD31-stained tumours grown in WT and *Tie2^PEKO* mice. (**c**) Quantification of vessel size of CD31-stained tumour vessels in *Tie2^PEKO* mice normalized to WT average. (**d**) Representative images of the tumour vasculature stained for CD31 and NG2 in tumours grown in WT and *Tie2^PEKO* mice. (**e**) Quantitative analysis of pericyte coverage in tumours grown in *Tie2^PEKO* mice normalized to WT average. (**f**) Representative images of tumour sections stained for CD31 24 h after tail-vein injection of rhodamine-labelled microspheres into WT and *Tie2^PEKO* mice. Arrows point towards microspheres inside and arrowheads outside of tumour vessels. (**g**) Quantification of microspheres per mm² in tumours grown in WT and *Tie2^PEKO* mice normalized to WT average. (**h**) Quantification of sphere area per total area in tumours grown in WT and *Tie2^PEKO* mice normalized to WT average. Scale bars: 100 µm (**a**,**d**), 50 µm/100 µm (**f** total/zoom). Data are shown as mean ± s.d. Statistics were performed using Mann–Whitney *U* test. *$P < 0.05$, **$P < 0.01$.

Ang2 on Tie2 (refs 45,46). Interestingly, the microarray experiments pursued in the present study had identified low levels of Tie2 expression in all analysed pericyte populations. Yet, none of the analysed pericyte populations expressed Tie1, which could explain why Ang2 would act strictly as an antagonist of Ang1/Tie2 signalling in pericytes.

Taken together, the present study expands the current concepts of vascular Ang/Tie signalling by establishing the contribution of pericyte Tie2 signalling to vascular maturation. Consequently, we conclude that the classical endotheliocentric view of Tie2 signalling with Ang1 acting in a paracrine manner and Ang2 through an autocrine loop needs to be revised in favour of a bi-directional reciprocal model in which the EC signalling is complemented reciprocally by an autocrine Ang1/Tie2 loop in pericytes and paracrine acting Ang2 (Fig. 7).

## Methods

**Mice.** *Tie2^PEKO* were generated by breeding *Ng2-Cre* mice (Jackson Laboratory, #008533)[39] with Tie2^fl/fl mice bred in the C57/Bl6 background[37]. For reporter analysis, *Ng2-Cre* mice were crossed to mT/mG reporter mice or Rosa-YPF^fl/fl mice, in which Cre-mediated excision resulted in GFP or YFP expression, respectively[36,47]. Animals were maintained on the C57BL/6 background and 8–10-week-old mice of both genders were used if not otherwise indicated. Mice were

housed in sterile cages, maintained in a temperature-controlled room and were fed autoclaved food and water. All animal experimentation was approved by the institutional and governmental Animal Care and Use Committees (permissions G-25/11 and G242/16 from the Regierungspräsidium Karlsruhe, Germany).

**Analysis of postnatal retinal angiogenesis.** Pups were killed at P4 or P6, eyeballs were isolated and fixed in either methanol at − 20 °C overnight or in 4% or 2% paraformaldehyde (PFA)/PBS for 1 h at RT. Isolated retinas were blocked and permeabilized with retina blocking buffer (10% normal goat serum, Dako, #X0907, in 0.5% Triton-X 100 and 1% BSA in PBS) for 1 h at RT. The retinal vasculature was stained with Alexa488- or FITC-conjugated Isolectin B4 (Sigma, 1:100) and NG2 antibody (Millipore, #Ab5320, 1:1,000) overnight at 4 °C. Subsequently, retinas were stained with the appropriate fluorescently labelled secondary antibodies and flat-mounted on microscope slides. Images were taken with the confocal microscopes Zeiss LSM710 and image analysis was accomplished with Fiji (ImageJ). The relative vessel area was calculated as Isolectin B4-positive area per retina area. Junctions and branches were counted in eight individual 600 µm × 600 µm fields of comparable regions in the retina and normalized to the vessel area. Pericyte coverage was determined by measuring the NG2-positive area associated with the vasculature and correlating it to the vessel area. If depicted as relative value, the single values per mouse were normalized to the average of the littermate WT or control treated animals. Each litter was analysed separately.

**Spheroid angiogenesis assay.** Female CB17 SCID mice (5–7 weeks) were purchased from Charles River (Sulzfeld #236). BP were transduced with lentivirus as described below before injection. Co-culture spheroids of ECs and pericytes were

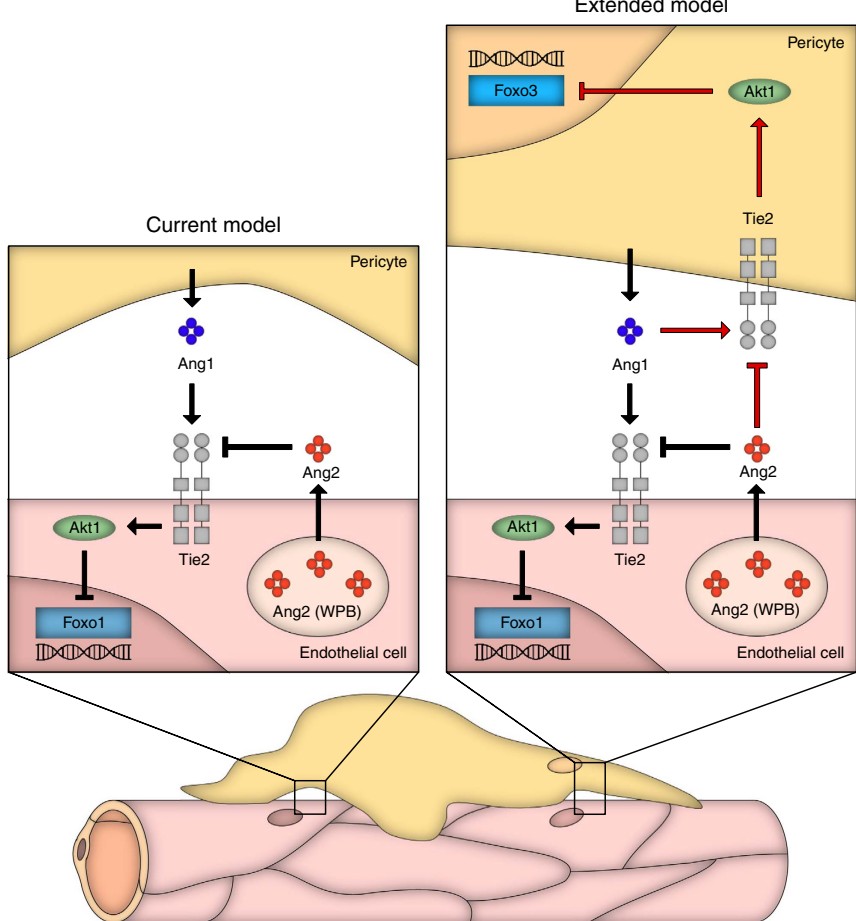

**Figure 7 | Model of vascular Ang/Tie signalling.** The schematic representation shows the established endothelial-centric model of Ang/Tie signalling (left) and the revised bi-directional, reciprocal EC-pericyte model of vascular Tie2 signalling as derived from the findings of this study (right; red routes show newly identified pathways). Vascular Ang/Tie2 signalling has so far exclusively been related to Tie2 expression by EC. Ang1 acts paracrine and is secreted by pericytes and other cells. Ang2 is almost exclusively expressed by EC and thereby acts as dynamic autocrine modulator of Ang1/Tie2 signalling. Extending this model, the findings of this study demonstrate that functional Tie2 receptors are also expressed by pericytes. Consequently, Ang1 expressed by pericytes (and likely other cells) activates pericyte Tie2 and contributes thereby to vascular maturation. Ang2 is produced by EC and acts paracrine on pericytes, thereby contributing to vascular destabilization through direct pericyte effects.

generated in a 1:1 ratio with a total of $3 \times 10^5$ injected cells as described below in the section 'in vitro sprouting assay'. After spheroid formation for 24 h, spheroids were collected by washing the plate with 10% fetal calf serum (FCS)/PBS and spun at 150g for 5 min. In the meantime, methocel/fibrinogen (Calbiochem, #341576)/ EC basal medium (PromoCell, #C-22210) mixture was prepared on ice in a 1:1:1 mixture. Pellets were washed carefully in 5 ml EC basal medium and centrifuged at 200g for 5 min. Spheroids were resuspended in 1 ml EC basal medium and collected by centrifugation at 200g for 5 min. In the meantime, 1,000 ng ml$^{-1}$ VEGFa (RELIATech, #300-036) and FGF2 (RELIATech, #300-003L) was added to the methocel/fibrinogen/EC basal medium mix. Next, 300 µl growth factor supplemented methocel/fibrinogen/EC basal medium mix were added to each spheroid sediment. Three hundred microliters Matrigel (BD Biosciences, #354230) and 4 µl of 1 unit µl$^{-1}$ thrombin (Calbiochem, #306190) were added and the solution was mixed with a syringe without a needle attached. The complete mixture was injected immediately with a 23G needle s.c. in the right flank[35,48]. Animals were kept for 21 days under appropriate conditions.

**Subcutaneous tumour models.** LLC or B16F0 cells ($1 \times 10^6$ cells) were injected s.c. into the right flank of *Tie2*$^{PEKO}$ and control mice. Mice were killed and tumours were collected at 13 or 10 days post tumour inoculation, respectively. For analysis of tumour growth over time, $1 \times 10^6$ B16F10 luciferase-positive cells were injected s.c. into the right flank of mice. Mice were anaesthetized with isoflurane three times a week and luciferase activity was measured 10 min following injection of 2 mg luciferin i.p. with a Xenogen IVIS imaging system (Perkin Elmer). Analysis of barrier function of tumour vessels was investigated as described previously[49]. Briefly, 15 µl of 100 nm diameter rhodamine-labelled fluorescent latex microsphere particles (LumaFluor Corp, Naples, USA) in 200 µl NaCl solution were injected i.v. 24 h before primary tumour removal. Analysis of vessel perfusion was performed

by i.v. injection of 60 mg kg$^{-1}$ body weight Hypoxy Probe (hpi, HP2-1000 kit) dissolved in 150 µl NaCl solution 45 min before tumour removal. Primary tumours were surgically removed under general anaesthesia. Mice with tumours not effectively removed or with subsequent tumour recurrence were removed from analyses. Mice were killed at the indicated time points or when they showed signs of ill health.

**Cell culture.** Cells were cultured at 37 °C, 5% CO$_2$ and 100% humidity. HUVEC were purchased from PromoCell (C-12203) and cultured in Endopan 3 medium (PAN Biotech, #P04-0010k)) supplemented with 3% FCS, the corresponding supplement mix and 1% penicillin/streptomycin. Human brain vascular pericytes (BP) were purchased from ScienCell (#1200) and cultured in pericyte medium (ScienCell, #1201) supplemented with 2% FCS, 1% of the corresponding pericyte growth supplement and 1% penicillin/streptomycin. Human placenta pericytes were purchased from PromoCell (C-12980) and human pancreas (PA), lung (LP) and muscle (MP) pericytes were kindly provided by Dr Bruno Peault (University of Edinburgh, UK) and cultured as described for BP. Dermal fibroblasts were purchased from PromoCell (C-12300) and cultured in Dulbecco's modified Eagle medium (DMEM) (Gibco) supplemented with 10% FCS and % penicillin/streptomycin. Mouse brain ECs (bEND3) and fibroblasts (NIH3T3) were kindly gifted by Dr Andreas Fischer and Dr Courtney König (German Cancer Research Center, Germany). bEND3 and NIH3T3 cells were cultured in DMEM GlutaMAX (Gibco) supplemented with 10% FCS and % penicillin/streptomycin. Mouse peritoneal macrophages (mΦ) were isolated from the peritoneal cavity of wild-type mice and cultured in RPMI (Gibco) supplemented with 10% FCS and % penicillin/streptomycin. For treatments with Ang1 or Ang2, cells were starved overnight with media containing 0.5% FCS. Before stimulation, medium was changed to basal medium. Cells were stimulated with the corresponding cytokine (Ang1 or Ang2, R&D,

400 ng ml$^{-1}$) for the indicated time points at 37 °C. LLC cells and B16 melanoma cells were kept in 10% FCS/1% penicillin/streptomycin/DMEM (Life Technologies).

**Transfection of cultured cells.** For siRNA-mediated Tie2 silencing, cells were transfected with two independent silencer select Tie2 siRNAs or control siRNA (Thermo Fisher, #s13983 and #s13984) using Oligofectamine transfection reagent (Life Technologies) in Opti-MEMI(1x) + GlutaMAX-I (Life Technologies) according to the manufacturer's instructions. Medium was exchanged after 4 h and gene and protein expression were analysed after 24, 48 or 72 h.

For lentiviral-based transduction, cells were washed with PBS prior to transfection. Lentivirus, introducing shTie2 (GE Dharmacon, #V2LHS_93160 and #V2LHS_232742), was added with an MOI (multiplicity of infection) ranging from 8 to 10 in fresh medium. Medium was replaced after incubation for 12 h at 37 °C. HUVEC were selected with 0.35 µg ml$^{-1}$ and BP with 0.7 µg ml$^{-1}$ puromycin 24 h after infection for an additional 72 h at 37 °C. Ang2 overexpression (Invitrogen, #V368-20) in HUVEC was performed similarly; however, kanamycin was used for selection. Supernatants were collected 24 and 48 h after transfection.

**Proliferation assay.** EdU incorporation and flow cytometric analysis were performed according to the manufacturer's instructions (Life Technologies, #C10634).

**Apoptosis assay.** Annexin V staining and flow cytometric analysis were performed according to the manufacturer's instructions (ebioscience, #88-8005-74).

**Scratch assay.** Cell culture chamber inserts (Ibidi) were placed in a 24-well tissue culture dishes. Subsequently, cells were seeded and grown overnight. After removal of the inserts, lateral migration of cells was visualized under the microscope. Pictures were taken at 0, 2, 4, 6, 8 and 10 h after removal of the chambers.

**In vitro sprouting assay.** HUVEC and BP were mixed 1:1 in Endopan 3 containing 20% Methocel (Sigma). Twenty-five microliters cell suspension drops were pipetted on non-adherent plastic plates[37]. Subsequently, plates were turned upside-down to form the so-called hanging drops in which spheroids are contained. Plates were incubated for 24 h at 37 °C and spheroids were collected by washing the plates with 10% FCS/PBS. Spheroids were centrifuged at 200g for 5 min and resuspended in 20% FCS and 80% methocel. The collagen mix was prepared on ice using Collagen type I (isolated from rat tail tendons), Medium 199 and NaOH (1 M) in an 8:1:1 ratio. Additionally, 1× HEPES buffer was added to the mix to adjust the pH. The collagen solution was mixed with the spheroid solution in a 1:1 ratio and transferred to a 24-well plate. For polymerization, gels were incubated for 30 min at 37 °C and then stimulated with 50 ng ml$^{-1}$ hVEGF (R&D) in basal EC medium. The assay was stopped after 24 h of incubation at 37 °C by adding 1 ml 10% PFA per well. Pictures of ten spheroids per gel were taken on an Olympus TH4–200 microscope and average sprout length was measured with Fiji.

**Cell isolation.** For EC and pericyte isolation from lungs via FACS, mice were killed and lungs were removed. Subsequently, lungs were digested with DMEM containing 1.25 mM CaCl2, 200 U ml$^{-1}$ collagenase I and 10 µg ml$^{-1}$ DNase I at 37 °C for 1 h. Single-cell suspensions of the digested organ were prepared by passing through 18G and 19G cannula syringes and filtering the lysates through a 100 µm cell strainer. Blood cells were depleted by staining for CD45-APC-Cy7 (BD Pharmingen, #567659, 1:500) and TERR119-APC-eFluoro780 (ebioscience, #47-5921, 1:200) for 30 min at 4 °C in PBS/5% FCS. After washing with PBS/5% FCS, stained cells were depleted by incubating the cell suspension with 500 µl Dynabeads magnetic beads (Life technologies, #1141D) for 30 min at 4 °C on a rotator. The remaining cells were stained with antibodies against CD31-PE-Cy7 (BD Pharmingen, #561410, 1:200) in PBS/5% FCS for 30 min at 4 °C. Dead cells were excluded by PI staining (Sigma, 1:3,000). CD45$^-$ TERR119$^-$ PI$^-$ CD31$^+$ EC cells and CD45$^-$ TERR119$^-$ PI$^-$ CD31$^-$ YFP$^+$ pericytes were sorted with a FACS Aria Cell Sorter (BD Biosciences). Pericytes from mouse brains were isolated from six pooled samples of the same genotype[50]. Mouse brains were dissected and chopped into small tissue pieces in MEM-HEPES (Sigma, #M7278). Following centrifugation for 5 min at 300g, tissue suspensions were resuspended in 5 ml dissociation solution containing MEM-HEPES, 135 units papain (Papain Dissociation System, Worthington Biochemical Corporation, #LK003176) and 615 units DNase I (Papain Dissociation System, Worthington Biochemical Corporation, #LK003170). After 1 h and 10 min of incubation at 37 °C, tissue was broken using first a 19G followed by a 21G needle. After trituration, the brain homogenate was mixed with 7 ml of 22% BSA and centrifuged at 500g for 10 min. Following centrifugation, the tube was tilted in a 45° angle to free the myelin at the top of the tube. The remaining blood vessel pellet was washed in 2 ml EC growth medium containing F12 Ham medium (Sigma, #N4888), 10% FCS, 1% penicillin/streptomycin, 30 µg ml$^{-1}$ EC growth supplement (Millipore, #02-102), 2.5 µg ml$^{-1}$ ascorbate (PAN Biotech, #P02-0010S4), 4 mM L-glutamine (PAA Laboratories, #M11-004) and 40 µg ml$^{-1}$ heparin (PAN Biotech, #P02-0010S8). Cells were spun at 300g for 5 min, resuspended in 8 ml EC growth medium and

plated in two wells of a six-well plate coated for 2 h at 37 °C with type I Collagen (isolated from rat tail tendons). The EC growth medium was replaced next morning by fresh medium. After cells had reached confluence, cells were split up to passage 5. Medium was changed to pericyte growth medium during the transition from passage 1 to passage 2 (iXCells, #MD-0030).

**Flow cytometry.** Surface expression of Tie2 in BP and HUVEC as well as isolated mBP was analysed by flow cytometry. Single cell suspensions were incubated with conjugated mouse-anti-Tie2-APC (R&D, #FAB3131A, 1:10) or rat-anti-Tie2-PE (ebioscience, #12-5987, 1:100) in basal medium containing 1% FCS for 30 min on ice. Corresponding isotype control, mouse IgG-APC (R&D, #IC002A, 1:10) rat IgG-PE (ebioscience, #12-4301-81, 1:100), was used. Sample acquisition was performed with a BD FACS Canto II flow cytometer and subsequent analysis was done using FlowJo software.

**Immunofluorescence stainings of cells and tissues.** Cultured cells were washed with PBS and fixed in 4% PFA/H₂O for 10 min or in Methanol/Acetone (1:1) for 2 min, washed and blocked with 10% goat serum and 3% BSA/PBS containing 0.3% Triton-X 100 for 30 min. Primary antibodies, Tie2 (R&D, #AF313, 1:100), Calpain1 (Sigma, #C5736, 1:100), FOXO3A (Cell Signaling, #2497, 1:100), actin (phalloidin, Invitrogen, #A12379, 1:500) NG2 (Millipore, #Ab5220, 1:100), CD31 (BD Bioscience, #553370, 1:100) and Mac-1 (BD Bioscience, #550282, 1:50) were prepared in 1% BSA/PBS containing 0.3% Triton-X 100 and incubated overnight at 4 °C. Cells were washed, incubated with the corresponding secondary antibody and Hoechst (Sigma, #33258, 1:1,000) for 30 min at RT in the dark and coverslips were mounted with Fluoromount G (Dako). Images were acquired with a LSM710 Zeiss confocal laser scanning system or Olympus IX81 light microscope. Analysis was performed using Zeiss LSM software or Olympus image analysis software (Cell-F).

Resected Matrigel plugs were fixed in zinc fixative overnight at 4 °C, dehydrated, embedded in paraffin and cut into 6 µm sections. For analysis of plug vascularization, CD34 (Novocastra, #NCL-END, 1:50) immunofluorescence staining was performed. Antigen-retrieval of the deparaffinized sections was accomplished with 8 µg ml$^{-1}$ Proteinase K in TE-buffer for 10 min at 37 °C. Sections were blocked with 10% normal goat serum for 1 h at RT and CD34 staining was performed overnight at 4 °C. Following washings, slides were incubated with the respective secondary antibody and Hoechst 1:5,000 at RT for 45 min and mounted with Fluoromount medium (eBioscience). Pictures of the entire plug area were taken using the Zeiss Cell Observer with the × 20 objective and image analysis was accomplished with Fiji. Six sections from different plug regions were analysed and the number of vessels and the vessel area were correlated to the plug area.

Resected tumours were embedded in Tissue-Tek OCT compound, frozen on dry ice and cut into 8 µm sections. Thirty sections in three different tumour regions in a distance of 800 µm to each other were prepared per tumour. Tumour cryosections were fixed with ice-cold methanol for 10 min at − 20 °C or with 4% PFA for 20 min at RT depending on the subsequent staining procedure. Following washing, sections were blocked with 10% ready-to-use goat serum (Invitrogen, #50062Z) for 1 h at RT. Staining with the primary antibodies against CD31 (BD Bioscience, #553370, 1:100; Biolegend, #102514, 10:100), NG2 (Cell Signaling, #4235, 1:1,000; Millipore, #Abs5320m 1:1,000), desmin (Abcam, #Ab15200-1, 1:200) or Hypoxy Probe Mab1-FITC (hpi, HP2-1000 kit 4.3.11.3,1:50) was done overnight at 4 °C. Subsequently, sections were stained with the appropriate fluorescently labelled secondary antibody and Hoechst dye (Invitrogen, #H3570, 1:2,000)/PBS for 1 h at RT and mounted with DAKO mounting medium. Pictures were taken as whole area tile scans using the Zeiss Cell Observer with the × 20 objective, fixed intensity and gain settings and image analysis was accomplished with Fiji. Three sections from three different tumour layers were analysed per staining. The average of number of vessels within the whole section and the vessel area were correlated to the total tumour area. Pericyte coverage was determined by normalizing the vessel associated NG2- or desmin-positive area to the total CD31 area. Hypoxic areas were normalized to the total tumour area. If not indicated differently, values were expressed as relative values normalizing each value to the average value of the control samples and thus allowing the comparison of independent experiments.

**Transmission electron microscopy of tumours.** Vibratome sections of sub-cutaneous tumorigenic tissue, pre-fixed in 4% formaldehyde/1% glutaraldehyde/1 mM MgCl₂/100 mM NaP buffer at pH 7.4 were post fixed with buffered 1% OsO4 following enbloc stain with 0.5% ethanolic uranl acetate and embedding in epoxide. Ultrathin sections of 60 nm nominal thickness and post-stained with uranyl and lead were investigated with an EM910 (Carl Zeiss, Germany) equipped with a CCD-camera (sharp-eye, TRS, Germany).

**ELISA.** Angiopoietin-2 protein levels of cell culture supernatants were measured by ELISA (R&D, #DANG20) according to the manufacturer's instructions. Samples were measured with a LUMIstar OPTIMA (Thermo Fisher) at 450 nm.

**Immunoprecipitation.** Cells were lysed on ice using lysis buffer (10 mM Tris-HCl pH 7.4, 150 mM NaCl, 1% Triton X-100, protease inhibitor mix G (Serva) and 2 mM Na3VO4). Cell lysates were incubated with protein G sepharose beads (GE Healthcare) and 3 µg Tie2 antibody (Millipore, clone Ab33, #05-584) overnight at 4 °C on a rotator. Beads were washed by centrifugation at 100g, 4 °C for 2 min with lysis buffer containing 2 mM Na₃VO₄ and boiled with 2 × protein sample buffer at 95 °C for 10 min. Immunoprecipitates and cell lysates were separated by SDS–PAGE followed by western blotting.

**Western blot analysis.** Cells were lysed in RIPA lysis buffer. Proteins were separated by SDS–PAGE and transferred on a nitrocellulose membrane. The membrane was blocked with 3% BSA/PBS for 1 h at RT followed by incubation with the corresponding primary antibody in 1% BSA/PBS-T overnight at 4 °C: pTyr (Millipore, #05-321, 1:500), Tie2 (R&D, #AF313, 1:1,000), Tie2 (R&D, #AF762), pAKT (Cell Signaling, #4060S, 1:1,000), AKT (Cell Signaling, #9272S, 1:1,000), FOXO1 (Cell Signaling, #9454, 1:1,000), pFOXO3A (Cell Signaling, #9466, 1:1,000), FOXO3A (Cell Signaling, #12829, 1:1,000), VEGFR2 (Cell Signaling, #2479, 1:1,000), tubulin (Sigma, #T8203, 1:5,000). Following washings, membranes were incubated with a specific horseradish peroxidase-conjugated secondary antibody (DAKO, 1:5,000) for 1 h at RT, washed and proteins were visualized by incubation with ECL (Thermo Fisher). Quantification was performed in Fiji. Uncropped western blots/arrays and larger blot/array areas are presented in Supplementary Figs 15–19.

**Proteome profiler assay.** Mouse angiogenesis and phospho-receptor tyrosine kinase array were performed according to the manufacturer's instructions (R&D, #ARY015 and #ARY014). A total amount of 225 µg protein, pooled from five whole tumour protein lysates of mice with the same genotype (75 µg each), was used. Human phospho-receptor tyrosine kinase array was performed according to the manufacturer's instructions (R&D, #ARY001B). A total amount of 198 µg protein, pooled from three independent experiments of the same siRNA transfection (66 µg each), was used. Quantification was performed in Fiji.

**Calpain activity assay.** BP were seeded two days before the start of the experiment. Medium was changed to starvation medium and incubated for 12 h. Cells were stimulated using 400 ng ml⁻¹ rhAng2 for 1 h. Cells were detached from the plate using accutase, counted and a total of 100,000 cells were suspended in 100 µl extraction buffer. Samples were incubated for 20 min on ice and mixed gently. After a 1 min centrifugation (10,000g) step, the supernatant was transferred into a fresh tube and placed on ice. The protein concentration was determined by Bradford protein measurement. Cell lysates (around 100 µg) were diluted to 85 µl of extraction buffer. As a positive control, 1 µl of active calpain was added to 85 µl extraction buffer. Calpain inhibitor (1 µl) added to treated cell lysate was used as a negative control. To each assay, 10 µl of 10 × Reaction Buffer and 5 µl of calpain substrate was added and incubated at 37 °C for 1 h in the dark. Samples were measured in a fluorometer equipped with a 400 nm excitation filter and 505 nm emission filter.

**RNA extraction and qPCR analysis.** RNA of FACS-sorted mouse cells were isolated with the Arcturus PicoPure RNA Isolation Kit (Life Technologies) according to the manufacturer's instructions. RNA of cells in culture was isolated with the RNeasy Mini Kit (RNeasy kit, Qiagen) according to the manufacturer's instructions. RNA from tumours was isolated by smashing the tissue pieces in TRIzol followed by Chloroform extraction. Afterwards, RNA was isolated with the RNeasy Mini Kit (RNeasy kit, Qiagen) according to the manufacturer's instructions. cDNA was synthesized with the QuantiTect Reverse Transcription Kit (Qiagen) according to the manufacturer's instructions. Gene expression analysis was performed by sqPCR or qRT–PCR. TaqMan reactions were performed on a StepOnePlus Real-Time PCR System (Applied Biosystems). The following primers were used for sqPCR: TEKF: 5′-CCAGGATGGCAGGGGCTCCA-3′, TEKR: 5′-GGTAGCGGCCAGCCAGAA GC-3′; PECAM1F: 5′-GTGAAGGTGCATGGCGTATC-3′, PECAM1R: 5′-CACAAA GTTCTCGTTGGAGGT-3′; CDH5F: 5′-CCTTCTTCACCCAGACCAAG-3′, CDH5R: 5′-GTGAAAGCGTCCTGGTAGTC-3′; CSPG4F: 5′-CCTGGAGAATGGT GGAAGAG-3′, CSPG4R: 5′-GGCCTGTGTTTGTAGTGAGGA-3′; CD248F: 5′-TC TGCCTGTGCTACCTTCTG-3′, CD248R: 5′-GCCTTTCTTGTCACCTCTGG-3′; GAPDHF: 5′-TGTCTCCTGCGACTTCAACA-3′, GAPDHR: 5′-GCCATGTAGGC CATGAGGT-3′. The following TaqMan Gene Expression assay (Applied Biosystems) Taqman probes (Life Technologies) were used: TEK (Hs00945146_m1), ANGPT2 (Hs01048042_m1), PDGFRb (Hs01019589_m1), Desmin (Hs00183740_m1), ACTA2 (Hs00426835_g1), CD248 (Hs00535586_s1), ANGPT1 (Hs00375822_m1), TIE1 (Hs00892696_m1), PECAM1 (Hs00169777_m1), CDH5 (Hs00901463_m1), Tek (Mm00443254_m1), Cd31 (Mm01242584_m1), Pdgfrb (Mm00435546_m1), Epha2 (Mm00438726_m1), Ephb2 (Mm01181021_m1), Efna1 (Mm01212795_m1), Efna4 (Mm00433013_m1), Efnb1 (Mm00438666_m1), Efnb2 (Mm00438670_m1), Vegfa (Mm01281449_m1), Vegfb (Mm00442102_m1), Vegfc (Mm00437310_m1), Vegfr1 (Mm00438980_m1), Vegfr2 (Mm01222421_m1), Vegfr3 (Mm01292604_m1). Uncropped gels and larger gel areas are presented in Supplementary Fig. 15.

**Microarray.** For gene expression analysis, microarrays were performed by the DKFZ Genomics and Proteomics core facility (Heidelberg, Germany). Briefly, RNA was isolated with the Arcturus PicoPure RNA Isolation Kit (Life Technologies) and RNA quantity and quality were checked using the Agilent RNA 6,000 Pico Kit on an Agilent Bioanalyzer. cDNA was hybridized on a HumanHT-12 Expression BeadChip Array (Illumina) according to the manufacturer's protocol. Microarray data were normalized and analysed with the Chipster software. Only genes with a significantly differential expression (P < 0.05) were considered for further analysis. The microarray data with the description of the experimental design are deposited under GEO number GSE86885.

**Statistical analysis.** All results are expressed as mean ± s.d. n represents the number of independent experiments or the number of mice analysed per group. Statistical analyses were performed using GraphPad Prism 6. Differences between experimental groups were analysed by two-tailed unpaired Student's t-test, non-parametric Mann–Whitney U test or one-way ANOVA with Dunnett post-test as indicated. P < 0.05 was considered as statistically significant (*P < 0.05, **P < 0.01, ***P < 0.001).

**Data availability.** All microarray data supporting the findings of this study have been deposited in the National Center for Biotechnology Information Gene Expression Omnibus (GEO) and are accessible through the GEO Series accession number GSE86885. The authors declare that the main data supporting the findings of this study are available within the article and its Supplementary Information files. Extra data are available from the corresponding author (H.G.A.) on request.

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

## Acknowledgements

The authors would like to acknowledge the excellent technical support of E. Besemfelder, M. Riedel, J. Wojtarowicz and C. Spegg, the DKFZ Laboratory Animal Facility (Prof K. Reifenberg), the DKFZ Light Microscopy Core Facility (Dr F. Bestvater), the DKFZ FACS Core Facility (Dr S. Schmitt), the DKFZ Electron Microscopy Core Facility (Dr K. Richter) and the DKFZ Genomics and Proteomics Core Facility (Prof S. Wiemann). This work was supported by grants from the SFB-TR23 'Vascular Differentiation and Remodeling' (project A3, to H.G.A.), the SFB873 'Maintenance and Differentiation of Stem Cells in Development and Disease' (project B6, to H.G.A.), and the Leducq Transatlantic Network of Excellence 'Lymph Vessels in Obesity and Cardiovascular Disease' (to H.G.A.).

## Author contributions

M.T., L.M. and H.G.A. conceived and designed the study; M.T., L.M. and A.H. developed methodology; M.T., L.M., A.H., L.S., N.G., S.S., T.R., Z.H., K.S., J.H., S.H., A.B. and K.S. performed experiments and acquired data; M.T., L.M., A.H., K.S. and H.G.A. analysed and interpreted data; M.T., L.M., K.S. and H.G.A. wrote the manuscript; all authors discussed the results and commented on the manuscript; K.S. and H.G.A. supervised the study.

## Additional information

**Competing interests:** The authors declare no competing financial interests.

