## [Peer Review File · Nature Communications]

Reviewers' comments:

Reviewer #1 (expert in Tie2 signaling and angiogenesis)

Remarks to the Author:

This is an important report that provides new insights into the role of Tie2 in pericytes, extending our understanding of Tie2's function in the vasculature beyond its effects in endothelial cells. The authors effectively show that functional Tie2 is expressed in pericytes. Loss of Tie2 in pericytes results in a modest delay in retinal vascular development (developmental angiogenesis) and a significant increase in tumor angiogenesis as a result of an increase in microvascular density and a loss of tumor vessel pericytes with increased vascular leak. Mechanistically, the data suggest that this is due to Ang1/Tie2-mediated phosphorylation of FOXO3A and inhibition of migration. Overall, the studies are well done and provide important new information in the Tie2 field, and my comments are mostly minor.

Major comments:

1. Effects of Ang1/Tie2 - The most compelling data are from the knockout mice, and the authors show that the phenotype of NG2-Tie2 KO mice have only a modest phenotype in the developmental context but a more marked phenotype in the context of tumor angiogenesis. The authors provide evidence that treatment with Ang2 mimics pericyte Tie2 loss-of-function. However, the paper might be strengthened by complementary evidence in support of an Ang1 gain-of-function effect. Data in Fig 6g,h show that Ang1 induces phosphorylation of FOXO3A. Does this have an effect on pericyte migration? Furthermore, the loss of pericyte numbers in the NG2-Tie2 KO tumors raises the question of whether Ang1/Tie2 signaling induces survival or proliferation of pericytes, either of which would be consistent with data in Fig 1i showing Ang1-mediated Akt activation in brain pericytes.

Minor comments:

1. Microspheres - Apparently the purpose of the microsphere injection in Fig 4 is to quantify vascular permeability, although the authors state it was to assess "functionality of tumour vessels". The authors should clarify this, as "functionality" could indicate perfusion, and microspheres can be used to that purpose. Accordingly, please clarify their size. The methods state "100 diameter" microspheres were used. Presumably they are 100 micrometers in diameter.

2. Fig 5c - Why was CD34 used instead of CD31 to stain ECs?

3. Suppl Fig 5d - Ang2 expression should be expressed in mass values, e.g., ng/ml.

Reviewer #2 (expert in tumor angiogenesis)

Remarks to the Author:

In this manuscript Teichert and co-workers demonstrated Tie2 regulates pericytes migration through Calpain/Talin/FAK and AKT/FOXO3A signaling cascades, which is a distinctive observation as conventionally Ang/Tie signaling has been known to be endothelial specific and has a significant pathophysiological role already shown by different laboratories including Dr. Augustin's lab. By using transgenic animal models as well as in vitro experiments, the authors have identified that the Tie2 expressed in pericyte playing a regulatory role in vessel maturation. They showed that mural cell specific Tie2 deletion causes a temporary delay in pericytes coverage in postnatal retina and also knocking down of pericyte specific Tie (Ng2-TieKO) mice showed increasing tumor growth may be due to more abnormal microvessel density, size and less pericytes coverage. Overall the manuscript

addressed an important role of Tie2 on pericytes' function and supported the current notion that Tie2 might not have solely endothelial specific role that was originally thought to be.

Comments:

- a) One major comment/suggestion is that the investigators proposed the novel hypothesis and proved their concept by the several biochemical and genetic models. However, one point need to be addressed is that whether this Tie2 expression in pericytes of any disease process. It will be of importance to show whether any disease process does this hypothesis still valid? This is a suggestion and by showing the expression pattern of Tie2, will be a string supportive data the work.
- b) In Fig 1a, which one is Tie2 column?
- c) The investigators should also show, at least in the supplemental data that shTie2 1 and 2 don't have any effect to other genes such as Tie1.
- d) In Fig 2 (e,f), the investigators claimed that the co-localization analysis of EC and pericytes revealed a significant reduction of NG2 coverage in Ng2-Tie2KO mice, however, the figure itself didn't show that significance difference. One option is that the investigator can enlarge some portion of the figure and showed the difference.
- e) In Fig 4, the authors elegantly showed that there are smaller blood vessels in the tumors of Tie (Ng2-TieKO) mice than that of control. It would be a great idea to show that there is no effect of VEGF expression on those tumor cells or VEGFR2 phosphorylation didn't alter.
- f) In Fig 6h, did the staining of the top panel was phospho-FOXO3A or just FOXO-3A?

Reviewer #3 (expert in tumor angiogenesis)

Remarks to the Author:

The authors describe a novel genetically manipulated mouse model where Tie2 has been deleted in NG-2-positive cells in vivo. The data are clear and in general the paper is well written. The manuscript describes the identification of Tie2 in different pericyte populations. This per se is not novel, and the authors state themselves that this has been shown before. However, what is novel is the demonstration that pericyte coverage of blood vessels is delayed in NG2Tie2fl/fl mice. Although how this fits into the story is not clear, it is novel. In addition they show that tumor growth and angiogenesis is enhanced in NG2Tie2fl/fl mice. In vitro angiogenesis was also enhanced when NG2-driven Tie2 was deleted. Together the results suggest that mural Tie2 acts to inhibits tumour angiogenesis. However the data presented seem as if they are beginning of a few different stories and the end of none. A robust molecular mechanism is not provided. This manuscript has the potential to be an important contribution, but it requires some significant amendments to make it acceptable.

Detailed comments include:

1. Figure 1b. CD248 is clearly expressed by the BPs but published evidence suggests that it is also expressed by fibroblasts. The authors should provide a profile of PC and fibroblast marker expression in their pericyte preparations and define what they call a pericyte.
2. Fig i-j: In response to silencing of TIE2, using shTIE2, the Western blot and densitometry show that p-AKT is stimulated in response to Ang1, albeit to a much lesser extent than the control, in the brain pericytes. This is not seen with the HUVECs. Have the authors tried other shTIE2 to get better knockdown.
3. Fig2b: Deletion of Tie2 in the Ng2xTie2KO is apparently statistically significant but the levels of Tie2 loss appear more in line with Heterozygous levels. N-numbers for the RNA data, indeed for all the data, should be provided. Deletion in the 'KO' is not apparent. Can the authors clarify and comment on

the poor level of deletion? The authors should show Tie2 protein expression in these cells compared to WT controls. Importantly, loss of Tie2 protein in pericytes in vivo is required.

4. Supp Fig 2b: In mTmG reporter mice a green signal should be apparent after excision by Cre? Why did the authors immunostain for GFP and why did they use isolectin to stain for blood vessels? According to initial publications, no immunostaining should be required. Why is the signal yellow? These data appear confused. What are we meant to be looking at here? Single color figures of endogenous color signal from the reporter and merges would make this more convincing.

5. Suppl Fig 3a – what were the genotypes of the breeding pairs of these mice. The ratios of WT;HET:KO do not appear to be in line with the expected Mendelian ratios as suggested in the text?

6. Fig2e – A reduction in NG2-positive cell coverage is not evident in the images, but this is implied in the quantitation. In Fig 2f the data here do not seem robust. There is a single outlier in the WT that appears to sway the data to a reduction in NG2 coverage in the KO. An increase in n-numbers here would be required to make these data are robust. Indeed, since the authors show that this phenotype recovers in adults, would it be worth looking at other timepoints, eg Postnatally at 3 days or 4 day and ask whether this phenotype is reproducible?

7. Given that the retina phenotype corrects itself by 8 weeks what is the relevance of this data in this paper. This is not clear.

8. Fig3 e – the authors state that the data from the log-transformed growth curve “curves diverged at early stages of tumour growth”. Essentially both the lines are the same up to 10days, showing almost identical tumour growth. After this timepoint the lines start to diverge.

9. Authors state “Fig3 g. shows profound gain of function phenotype in mice with targeted deletion “. This statement appears to be an exaggeration. Importantly, the luciferase data are not correctly interpreted. The degree of luciferase signal depends on the delivery of luciferin to the tumour. Since the authors go on to show that the tumours in the KO mice have significantly higher numbers of blood vessels, the increase in luciferase signal could simply be a reflection of an increase in number of blood vessels. For longitudinal tumour growth studies these experiments should be repeated and ultrasound and/or caliper measurements used to determine the tumour growth over time.

10. Fig 4: Authors should explain why blood vessels have a small diameter with reduced PC coverage? It would be expected that reduced pericyte association would be associated with an increase in blood vessel diameter.

11. Fig 4 c: It is unclear what ‘vessel size(fold)’ measurements actually mean and how they were calculated is also unclear. Raw data for at least 40 vessels/ mouse and at least 6 mice per genotype should be shown. As it stands the images do not appear to reflect the quantitation on vessel size provided.

12. Fig 4 f-h: The authors suggest that the increase in blood vessel density is sufficient to enhance the tumour growth in the KO mice. However, given the reduction in vessel diameter what evidence do the authors have that the vessels are functional and not just leaky. The microsphere data images are very poor and it is almost impossible to count vessels that contain spheres. These should be replaced with better quality images. Indeed these data rather show leakage of spheres into the tumour mass and do not reflect actual functionality of the vessels. Functional perfusion studies using tail vein administered PE-PECAM-ab would determine whether the KO vessels, despite their smaller size, are well perfused or not. What is the value of the data in Fig h . A 1% difference seems hardly biologically significant.

13. Fig 4e – Again how was this measured. Is this the number of NG2-positive cells/endothelial cell , as the axis label suggests, or the number of pericytes associated with individual tumour blood vessels. If it is the latter the data should be given as raw data, not fold change. There is one outlier in data – if removed does it affect the statistics? Is this data point skewing the data to make a significance that may not be there if more samples were added to the experiment.

14. Fig 5: Evidence for equal numbers of pericytes in these assays needs to be confirmed histologically. The data should be supported by aortic ring data from the WT and KO mice. Why did the authors not simply carry out in vivo matrigel plug assays? Which growth factors are involved here? Growth factor induced angiogenesis assays would be much more informative and would help to develop a more robust mechanism.

15. Fig 6 –The cell confluency looks very different in shCtr versus shTIE2 I and II, could this reflect the wound closure results?

16. Densitometric analysis of Fig6d is required for repeat experiments.

17. Fig 6 e-h – it is really unclear how this potential molecular mechanism hangs together. Overall the mechanism of reduced migration of Tie2kd pericytes is not clear and how this relates to the in vivo phenotype is not addressed except hypothetically. A substantial improvement in the molecular mechanism is required.

18. Fig6g – nuclear and cytoplasmic fractions of FOXO3A with appropriate controls would be more informative

19. Since the authors identify a role for pericyte Tie2 in the control of regulators of invasion have the authors tried metastasis experiments; esp. as they see an increase in microsphere in tumour in KO mice.

20. Authors should explain more the relevance of including VECADTie2 embryo data? This seems to be an aside and nothing to do with the pericyte MS and perhaps not that novel.

21. The size of the FAK band in Fig 6d is incorrect. The bands highlighted are less than 100kDa based on the ladder (sup fig 9 for fig 6d). FAK should be approximately 125kDa. The bands on the gel at around 130kd are more likely to be the FAK band and here FAK appears to go up in the pAng2 lane and down in the shAng2 lane thus changing the interpretation of the molecular mechanism of how Tie2 controls pericyte behavior. The authors are advised to revisit these data as they don't seem robust as they stand.

Response to Reviewer's comments

Introductory remark

In revising the manuscript, we have opted to restructure the flow of the figures. The original manuscript validated that cultured pericytes express functional Tie2 (Fig. 1). We then moved on to show that genetic deletion of Tie2 in pericytes *in vivo* affects physiological (retina) and pathological (tumour) vessel maturation (Fig. 2-4). We then switched again to cellular experiments (Fig. 5 and 6). In the revised manuscript, we first establish the cellular phenotype of pericyte Tie2 and build on this the *in vivo* experiments. Also in line of the specific comments of the reviewers, we believe that this revised structure avoids jumping back and forth from *in vitro* and *in vivo* and has given the manuscript more coherences and better readability.

Reviewer #1

GENERAL COMMENT: This is an important report that provides new insights into the role of Tie2 in pericytes, extending our understanding of Tie2's function in the vasculature beyond its effects in endothelial cells. The authors effectively show that functional Tie2 is expressed in pericytes. Loss of Tie2 in pericytes results in a modest delay in retinal vascular development (developmental angiogenesis) and a significant increase in tumor angiogenesis as a result of an increase in microvascular density and a loss of tumor vessel pericytes with increased vascular leak. Mechanistically, the data suggest that this is due to Ang1/Tie2-mediated phosphorylation of FOXO3A and inhibition of migration. Overall, the studies are well done and provide important new information in the Tie2 field, and my comments are mostly minor.

RESPONSE TO GENERAL COMMENT: We sincerely appreciate the overall positive assessment of our work and would like to thank the reviewer for his/her constructive comments. We have addressed all of the reviewer's suggestions as detailed below.

COMMENT 1: Effects of Ang1/Tie2 - The most compelling data are from the knockout mice, and the authors show that the phenotype of NG2-Tie2 KO mice have only a modest phenotype in the developmental context but a more marked phenotype in the context of tumor angiogenesis. The authors provide evidence that treatment with Ang2 mimics pericyte Tie2 loss-of-function. However, the paper might be strengthened by complementary evidence in support of an Ang1 gain-of-function effect. Data in Fig 6g,h show that Ang1 induces phosphorylation of FOXO3A. Does this have an effect on pericyte migration? Furthermore, the loss of pericyte numbers in the NG2-Tie2 KO tumors raises the question of whether Ang1/Tie2 signalling induces survival or proliferation of pericytes, either of which would be consistent with data in Fig 1i showing Ang1-mediated Akt activation in brain pericytes.

RESPONSE 1: We fully agree with the reviewer and would kindly ask him/her to consider our rationale for focussing on Tie2 and Ang2 as follows:

(i) Modest phenotype in postnatal developmental retinal angiogenesis and marked phenotype in tumour experiments: We consider this not a limitation of the study, but rather reflecting its beauty. Generally speaking, long-term analyses of nonlethal phenotypes more likely reflect mechanisms of adaptation and compensation rather than the consequences of the loss of a molecule. As such, the default of retinal angiogenesis is a fully mature vasculature. Thus, when knocking out Tie2 in pericytes, we delay angiogenesis and vessel maturation, but we don't compromise the endpoint (which is accomplished by hitherto unknown adaptation/compensation mechanisms). In contrast, in a tumour setting, in which the endpoint a chronically angiogenic and immature vasculature, the absence of Tie2 on pericytes yields a remarkable increase of tumour growth. Yet, in this experiment too, long-term analysis revealed that tumour growth rates (i.e., doubling times) at later stages of tumour growth are not altered (see also comment 8 of reviewer 3).

(ii) Ang1 gain-of-function: Following the reviewer's suggestion, we have performed Ang1 gain-of-function experiments. Mitogenic and motogenic effects of Ang1 on EC have been reported (1,2). Yet, generally speaking, Ang1 is a poor mitogen and motogen. The primary downstream effect is on PI3K/AKT-mediated survival signalling. In line with these well-established data for endothelial cells, Ang1 stimulation experiments in pericytes revealed a

robust anti-apoptotic effect as well as subtle, non-significant effects on proliferation and migration. The results of these additional experiments have been incorporated into the manuscript (**Supplementary Fig. 3**).

COMMENT 2: Microspheres - Apparently the purpose of the microsphere injection in Fig 4 was to quantify vascular permeability, although the authors state it was to assess "functionality of tumour vessels". The authors should clarify this, as "functionality" could indicate perfusion, and microspheres can be used to that purpose. Accordingly, please clarify their size. The methods state "100 diameter" microspheres were used. Presumably they are 100 micrometers in diameter.

RESPONSE 2: We apologize for omitting to mention the microsphere diameter in the methods section. This has been remedied in the revised manuscript. The microspheres are 100 nm in size. Their detection reflects primarily permeability and secondarily vessel perfusion. We consider these findings important because we had expected a less well perfused vasculature in the more angiogenic and more immature vasculature of tumours grown in *Tie2^{PEKO}*. To further validate perfusion, we have performed transmission electron microscopy (TEM) experiments showing erythrocytes throughout the microvasculature of tumours grown in WT as well as in the smaller diameter microvessels of *Tie2^{PEKO}* mice. These findings have been incorporated in the revised manuscript (**Supplementary Fig. 12a**). Moreover, we analysed tissue hypoxia by intravenous injection of Hypoxy Probe. These experiments identified a similar extend of hypoxic regions within the tumours if both genotypes (included into the revised manuscript as **Supplementary Fig. 12b**).

COMMENT 3: Fig 5c - Why was CD34 used instead of CD31 to stain ECs?

RESPONSE 3: Figure 3c (previously Figure 5c) shows the results of a spheroid-based grafting assay of co-culture spheroids consisting of shTie2 silenced human brain pericytes and human umbilical vein endothelial cells implanted into CB17 SCID mice. We have established this assay 9 years ago (3). Given the chimeric situation of the assay (human cells in immunocompromised mice), we established at the time several protocols to most robustly trace the grafted human cells in the host mice. CD34 turned out to be the most specific and least cross species-specific marker to trace the grafted human EC.

COMMENT 4: Suppl Fig 5d – Ang2 expression should be expressed in mass values, e.g., ng/ml.

RESPONSE 4: We sincerely appreciate this suggestion. Following the reviewer's advice, Ang2 levels are now expressed in ng/ml (**Supplementary Fig. 4d** of the revised manuscript).

Reviewer #2

GENERAL COMMENT: In this manuscript Teichert and co-workers demonstrated Tie2 regulates pericytes migration through Calpain/Talin/FAK and AKT/FOXO3A signalling cascades, which is a distinctive observation as conventionally Ang/Tie signaling has been known to be endothelial specific and has a significant pathophysiological role already shown by different laboratories including Dr. Augustin's lab. By using transgenic animal models as well as in vitro experiments, the authors have identified that the Tie2 expressed in pericyte playing a regulatory role in vessel maturation. They showed that mural cell specific Tie2 deletion causes a temporary delay in pericytes coverage in postnatal retina and also knocking down of pericyte specific Tie (*Ng2-TieKO*) mice showed increasing tumor growth may be due to more abnormal microvessel density, size and less pericytes coverage. Overall the manuscript addressed an important role of Tie2 on pericytes' function and supported the current notion that Tie2 might not have solely endothelial specific role that was originally thought to be.

RESPONSE TO GENERAL COMMENT: We sincerely appreciate the overall positive assessment of our work and would like to thank the reviewer for his/her constructive comments. We have addressed all of the reviewer's suggestions as detailed below.

COMMENT 1: One major comment/suggestion is that the investigators proposed the novel hypothesis and proved their concept by the several biochemical and genetic models. However, one point need to be addressed is that whether this Tie2 expression in pericytes of any disease process. It will be of importance to show whether any

disease process does this hypothesis still valid? This is a suggestion and by showing the expression pattern of Tie2, will be a string supportive data the work.

RESPONSE 1: The role of Tie2 on pericytes *in vivo* has been addressed in the present study in two independent tumour models, which are disease models. Obviously, we would like to follow the reviewer's suggestion to validate these preclinical findings by expression profiling in relevant human tumour samples. Yet, this is technically challenging and a little beyond what is presently doable. EC and pericytes are in very close spatial contact. It is in principle possible to dissociate EC and pericyte in high-resolution double stainings. For example, we have been able to show that the molecule identified by the Vogelstein group as tumour endothelial marker TEM1 (4) is in fact not expressed by EC, but rather by pericytes (5). Yet, Tie2 is much more abundantly expressed by EC compared to pericytes. The genetic approach of the present study unambiguously establishes a functional role of the low level Tie2 expression of pericytes. Yet, this does not imply that the molecule would be easy to trace in pericytes.

Despite these limitations, we have followed the reviewer's advice and established triple staining techniques to stain for Tie2 together with CD31 (EC) and NG2 (pericytes). We employed towards this end a broad panel of available reagents [R&D #AF313; Atlas Antibodies #HPA011738; Santa Cruz #sc-324; Millipore #05-584; R&D #AF2720-SP; R&D #AF313; Santa Cruz #sc-1616; ebioscience #14-5987-81; ebioscience #12-5987]. As shown in Fig. A below, we could successfully do the proof-of-principle to detect NG2/Tie2 co-staining. Yet, generally speaking the close proximity of EC and pericytes and the strong EC staining for Tie2 makes it really not realistic to pursue this for extensive expression profiling in tumours. Nevertheless, we believe that the genetic approach pursued in the present study provides strong evidence that pericyte Tie2 contributes to the regulation of tumour angiogenesis.

Figure A: Staining for pericyte mouse Tie2. Staining for CD31, NG2 and Tie2 in mouse tumours. Arrowheads point towards Tie2-NG2 co-localization. Scale bar = 10 μ m.

COMMENT 2: In Fig 1a, which one is Tie2 column?

RESPONSE 2: TEK in the first lane of Fig. 1a represents the name of the Tie2 gene. This has been indicated in the revised figure.

COMMENT 3: The investigators should also show, at least in the supplemental data that shTie2 1 and 2 don't have any effect to other genes such as Tie1.

RESPONSE 3: We sincerely appreciate the reviewer's comment and have performed further validation experiments using EC and brain pericytes following knockdown of Tie2. For pericytes, none of the analysed genes that are typically expressed by pericytes changed significantly upon knockdown (included as **Supplementary Fig. 1c, d** in the revised manuscript). Moreover, the reviewer asked for effects on *TIE1*. However, pericytes do not express *TIE1* (Fig. 1a [which is actually quite important and subject of another on-going study]). Interestingly, silencing of Tie2 in HUVEC did not change the expression of major EC genes such as *PECAM1*, *CDH5* or *ANGPT2* (**Supplementary Fig. 1d**). Yet, the expression of *TIE1* was significantly reduced with one of the shRNAs. However, shRNA-mediated Tie2 knockdown was only performed (Fig. 1 of manuscript) to compare previously published data of EC with pericytes. All further studies exclusively concentrated on Tie2 in pericytes. Thus, the potential Tie1 off target effect with one of the shRNAs has no consequences for any of the functional pericyte effects.

COMMENT 4: In Fig 2 (e,f), the investigators claimed that the co-localization analysis of EC and pericytes revealed a significant reduction of NG2-coverage in Ng2-Tie2KO mice. However, the figure itself didn't show that significance difference. One option is that the investigator can enlarge some portion of the figure and showed the difference.

RESPONSE 4: We apologize for including in the original manuscript images that were not perfectly representative to match the quantitation. We have replaced the images to better illustrate the difference in NG2 coverage (Figure 4e of the revised manuscript).

COMMENT 5: In Fig 4, the authors elegantly showed that there are smaller blood vessels in the tumors of Tie (Ng2-TieKO) mice than that of control. It would be a great idea to show that there is no effect on VEGF expression on those tumor cells or VEGFR2 phosphorylation didn't alter.

RESPONSE 5: We sincerely appreciate the reviewer's suggestion and followed his/her advice to investigate Vegf expression as well as VEGFR2 phosphorylation. Performing a mouse angiogenesis as well as a phospho-receptor tyrosine kinase proteome profiler assay of whole tumour protein lysates from WT and *Tie2^{PEKO}* mice did not reveal significant changes in VEGF ligand as well as VEGF receptor phosphorylation levels, including VEGFR2. Total protein levels of VEGFR2 in tumour lysates did not change upon Tie2 deletion in pericytes. Furthermore, gene expression analysis of Vegf ligands and receptors in whole tumour lysates did not show significant differences between knockout and wildtype samples. In summary, pericyte Tie2 loss does not interfere with VEGF signalling. The results of these experiments have been incorporated into the revised manuscript (**Supplementary Fig. 11**).

COMMENT 6: In Fig 6h, did the staining of the top panel was phospho-FOXO3A or just FOXO3A?

RESPONSE 6: Indeed, Fig. 2g (previously Fig. 6h) shows total FOXO3A reflecting absolute protein content in both compartments. Phosphorylation of FOXO3A leads to inactivation and causes its sequestration into the cytoplasm. Relocalization to the nucleus is achieved via dephosphorylation by protein phosphatases (6). As such, total FOXO3A staining and its differential localization pattern in the cytoplasm and in the nucleus upon Ang1 and Ang2 treatment reveals changes in the FOXO3A activation status following angiopoietin stimulation.

Reviewer #3

GENERAL COMMENT: The authors describe a novel genetically manipulated mouse model where Tie2 has been deleted in NG-2-positive cells in vivo. The data are clear and in general the paper is well written. The manuscript describes the identification of Tie2 in different pericyte populations. This per se is not novel, and the authors state themselves that this has been shown before. However, what is novel is the demonstration that pericyte coverage of blood vessels is delayed in NG2Tie2fl/fl mice. Although how this fits into the story is not clear, it is novel. In addition they show that tumor growth and angiogenesis is enhanced in NG2Tie2fl/fl mice. In vitro angiogenesis was also enhanced when NG2-driven Tie2 was deleted. Together the results suggest that mural Tie2 acts to inhibit tumour angiogenesis. However the data presented seem as if they are beginning of a few different stories and the end of none. A robust molecular mechanism is not provided. This manuscript has the potential to be an important contribution, but it requires some significant amendments to make it acceptable.

RESPONSE TO GENERAL COMMENT: We sincerely appreciate the overall positive assessment of our work and would like to thank the reviewer for his/her many constructive and thoughtful comments. We have performed a concerted effort to address all of the reviewer's suggestions as outlined below.

COMMENT 1: Figure 1b. CD248 is clearly expressed by the BPs but published evidence suggests that it is also expressed by fibroblasts. The authors should provide a profile of PC and fibroblast marker expression in their pericyte preparations and define what they call a pericyte.

RESPONSE 1: Human umbilical vein endothelial cells (HUVEC), brain and placenta pericytes (BP, PP) and dermal fibroblasts (Fib) were purchased from PromoCell (HUVEC #C-12203; PP #C-12980; Fib #C-12300) and ScienCell (BP #1200). Isolation procedures and analysed marker expression are reviewed in detail on the company's webpages. Pericytes from pancreas, lung and muscle were kindly provided by Dr. Bruno Peault (University of Edinburgh, UK) and their isolation has been published previously (7-9).

Beyond the published findings, we have further validated the brain pericytes (BP) that were used in most of the functional experiments. Microarray-based analysis validated the clear separation of different human pericytes from EC. We agree with the reviewer that a clear distinction of pericytes from fibroblasts is more difficult since there is a

substantial overlap in mesenchymal marker expression (10) and also increasing evidence for a lineage relationship between pericytes and fibroblasts (e.g. Ref. 11). Fibroblasts were also devoid of endothelial markers (including *Tie2*), but expressed *ACTA2* (α SMA), *PDGFRb* and *CD248* (endosialin). Yet, fibroblasts expressed rather low levels of *CSPG4* (NG2) compared to pericytes, which is one of the reasons why we employed NG2-Cre as driver in the genetic experiments of the present study. The results of these additional expression-profiling experiments are included in the manuscript (**Supplementary Fig. 1**) and validate the pericyte nature of the manuscript's functional and genetic experiments.

COMMENT 2: Fig 1i-j: In response to silencing of TIE2, using shTIE2, the Western blot and densitometry show that p-AKT is stimulated in response to Ang1, albeit to a much lesser extent than the control, in the brain pericytes. This is not seen with the HUVECs. Have the authors tried other shTIE2 to get better knockdown.

RESPONSE 2: We achieved a knockdown efficacy of >75% resulting in reduced p-Akt upon Ang1 stimulation. The scaling of Figure 1j and l might be misleading since the increase of p-Akt in HUVEC is in general much higher giving the impression that there is no residual p-Akt at all. Yet, the ratio of p-Akt levels remaining after Tie2 knockdown to shCtr following Ang1 stimulation are in fact rather similar in brain pericytes and endothelial cells taking the scaling of the axis into consideration. It is important to note that Tie2 levels in pericytes and correspondingly pTie2 are at a substantially lower level in pericytes compared to EC. Yet, the cellular and the genetic experiments establish quite unambiguously that this is functionally relevant.

COMMENT 3: Fig2b: Deletion of Tie2 in the Ng2xTie2KO is apparently statistically significant but the levels of Tie2 loss appear more in line with heterozygous levels. N-numbers for the RNA data, indeed for all the data, should be provided. Deletion in the 'KO' is not apparent. Can the authors clarify and comment on the poor level of deletion? The authors should show Tie2 protein expression in these cells compared to WT controls. Importantly, loss of Tie2 protein in pericytes *in vivo* is required.

RESPONSE 3: Regarding RNA data, n-numbers are included in all corresponding figure legends of the revised manuscript. Triplicates were measured for Figure 4b (previously Fig. 2b). However, six mice of the same genotype were pooled per sample to retrieve a sufficient amount of RNA for quantification.

Other than that, the reviewer touches on a very critical point of the study. Our group is the first to have generated floxed Tie2 mice and we have obtained extensive experience in the genetic deletion of Tie2 in different cell types. Generally speaking, the Tie2 locus is not trivial to target. For example, when targeting Tie2 in EC in adult mice (using VE-Cad-CreERT2), we get high recombination in tumour EC and hardly any recombination in normal organ endothelia (we get better recombination efficacy when using Pdgfb-CreERT2). We don't fully understand what drives recombination efficacy, but it is clearly not just the driver that is decisive, but also the locus. This is one of the reasons, why we are using for non-EC cells a constitutive rather than an inducible Cre.

We have performed very meticulous and genetically controlled cell sorting. By using *NG2-Cre* driven YFP recombination, we first ensure that our genetic analysis is not compromised by Cre leakage into the EC compartment (Figure 4a). This was most important to unambiguously ascribe the observed phenotype to the pericyte and not to the EC compartment. Concerning recombination efficacy in the pericyte compartment, we routinely yield between 50% and 60% (as shown in Figure 4b). We agree with the reviewer that this is not 100%. Yet, this is not a heterozygous value, but, of course, a mosaic situation. Importantly though, it is sufficient to trigger a phenotype. Thus, if anything, we are underestimating the functional role of pericyte Tie2. To clearly make this important point, we have briefly mentioned this in the discussion.

With regards to Tie2 protein detection upon recombination in pericytes *in vivo*, currently available methods reach their limitations. YFP-based sorting as it was done for the RNA data does not yield sufficient amounts of high quality protein to perform a Tie2 Western blot even when pooling multiple mice. Staining for Tie2 in mouse tumour sections and retinas can be performed successfully, but Tie2 molecules translocate to intercellular cell-cell contacts (12). Thus, it is not really possible to reliably distinguish EC- and pericyte-expressed Tie2 due to the close proximity and the formation of cell-cell contact between these two cell types (see also comment 1 of reviewer 2). Thus, we decided to isolate brain pericytes from WT and *Tie2^{PEKO}* mice. Pericytes were isolated as described previously (13). Six animals per genotype were pooled for successful cell isolation. Tie2 Western blot analysis confirmed the RNA

analysis and revealed an approximately 50% reduction of Tie2 protein expression in isolated pericytes. The isolated cells were negative for markers of other cell types reported to express Tie2 (CD31 for EC and Mac-1 for macrophages). The findings of these further experiments have been incorporated into the manuscript (**Supplementary Fig. 7**).

COMMENT 4: Supp Fig 2b: In mTmG reporter mice a green signal should be apparent after excision by Cre? Why did the authors immunostain for GFP and why did they use isolectin to stain for blood vessels? According to initial publications, no immunostaining should be required. Why is the signal yellow? These data appear confused. What are we meant to be looking at here? Single color figures of endogenous color signal from the reporter and merges would make this more convincing.

RESPONSE 4: We sincerely appreciate that the reviewer points out that mT/mG mice should show fluorescence without the need of further staining. However, staining for the gene product rather than the molecule's inherent fluorescence is nowadays widely used for better sensitivity. We have stained blood vessels with isolectin B4 and the pericyte signal was enhanced with a GFP staining enabling us to visualize the vasculature and vessel covering mural cells simultaneously. We have included into the revised manuscript the unambiguous single channel images in addition to the merged images (**Supplementary Fig. 6**).

COMMENT 5: Suppl Fig 3a – what were the genotypes of the breeding pairs of these mice. The ratios of WT;HET:KO do not appear to be in line with the expected Mendelian ratios as suggested in the text?

RESPONSE 5: Perfect Mendelian ratios oftentimes require an extensive experimental n that is mostly not achieved in double mutagenesis experiments. **Supplementary Fig. 8** (previously Supplementary Fig. 3a) shows the resulting litter genotypes of breedings heterozygous Tie2 mice. In general, Tie2 mice are bred homozygous on the Tie2 allele and, thus, the number of available breeding pairs for analysis is inherently limited. In revising the manuscript, we have expanded the experimental n. This is still not a perfect Mendelian ratio, but sufficient to support the statement made in manuscript: “*Tie2*^{PEKO} mice were born close to the predicted Mendelian ratio”. Importantly, if anything, the KO group is overrepresented and not underrepresented.

COMMENT 6: Fig2e – A reduction in NG2-positive cell coverage is not evident in the images, but this is implied in the quantitation. In Fig 2f the data here do not seem robust. There is a single outlier in the WT that appears to sway the data to a reduction in NG2 coverage in the KO. An increase in n-numbers here would be required to make these data are robust. Indeed, since the authors show that this phenotype recovers in adults, would it be worth looking at other timepoints, eg postnatally at 3 days or 4 day and ask whether this phenotype is reproducible?

RESPONSE 6: We sincerely appreciate the reviewer's comments and have extensively re-analysed all data. Importantly, there is a robust functional vessel maturation phenotype resulting in higher microvessel density in the *Tie2*^{PEKO} mice during the active phase of angiogenesis. As suggested by the reviewer, we have expanded the analysis to P4 retinas (peak lateral expansion). There is a significant reduction in MVD at P4, but at this stage not a reduction of pericyte recruitment (**Fig. 4c of revised manuscript**). MVD catches up at P6, when successively we observe a delay in pericyte recruitment (**Fig. 4e of revised manuscript**). We agree with the reviewer that it appears that this difference could be caused by an outlier. Yet, we performed both parametric (Student t test) and non-parametric (Mann-Whitney U test) statistics and the result is the same that there is a significant difference in pericyte recruitment on day 6. Together, this staggered phenotype is in line with our interpretation of a delay in vascularization and maturation during physiological developmental angiogenesis – without affecting the end point. Interestingly, in the tumour setting with uncontrolled delivery of angiogenic growth factors, loss of pericyte Tie2 strongly enhances MVD and correspondingly tumour growth. Following the reviewer's suggestion, we have also re-analysed the tumour data. Indeed, while the MVD phenotype is very robust, the single outlier in Figure 6e drives the p value beyond 0.05 when applying proper nonparametric statistical analysis. This has been corrected in the revised manuscript and the correct p value is given.

COMMENT 7: Given that the retina phenotype corrects itself by 8 weeks what is the relevance of this data in this paper. This is not clear.

RESPONSE 7: Many non-lethal KO show transient phenotypes or phenotypes upon challenge. The physiological interpretation of such observations is most appropriately that long-term non-lethal phenotypes reflect more the adaptation or compensation of the loss of a specific gene rather than the gene's actual function. As such, the default of postnatal retinal angiogenesis is full vascularization with a mature vasculature. Pericyte Tie2 KO is not lethal, but leads to a transient phenotype that in the long run is compensated/adapted to the default state (by hitherto unknown mechanisms). In contrast, the default of a tumour is a chronic and non-mature vasculature. Our interpretation is therefore that we see for exactly these reasons a sustained phenotype in the pathological situation of a tumour, but not in the physiological retinal setting (see also comment 1 of reviewer 1).

COMMENT 8: Fig3 e – the authors state that the data from the log-transformed growth curve “curves diverged at early stages of tumour growth”. Essentially both the lines are the same up to 10 days, showing almost identical tumour growth. After this time point the lines start to diverge.

RESPONSE 8: The reviewer touches on an important point of the study. We apologize for not having been able to properly phrase this in the original manuscript and have carefully reworded this in the revised manuscript: Many/most tumour studies focus on endpoints and ignore that fact that tumour growth needs to be considered kinetically over time. Given the exponential character of tumour growth curves, differences at early stages of tumour growth will increase over time simply as an epiphenomenon of the exponential growth. Log transformed data show tumour growth rates, i.e., doubling times. Obviously, such curves do not look as impressive as linear growth curves, but they more appropriately reflect the biological behaviour of the tumour. This is shown in **Figures 5g and 5f of the revised manuscript**. Figure 5g shows the typical exponential growth seen in many tumour studies. Figure 5f is a log transformed re-plot of the same data showing that tumour growth rates diverge between days 4 and 8 after which the difference in growth rates becomes smaller to show parallel growth rates (i.e., doubling times) between days 10 and 12. This formally proves that the phenotype is transient (as in the retina experiment); yet, it is sustained because the tumours grown in KO mice do not catch up. We consider this fully in line with our interpretation of the *Tie2*^{PEKO} phenotype.

COMMENT 9: Authors state “Fig3 g. shows profound gain of function phenotype in mice with targeted deletion “. This statement appears to be an exaggeration. Importantly, the luciferase data are not correctly interpreted. The degree of luciferase signal depends on the delivery of luciferin to the tumour. Since the authors go on to show that the tumours in the KO mice have significantly higher numbers of blood vessels, the increase in luciferase signal could simply be a reflection of an increase in number of blood vessels. For longitudinal tumour growth studies these experiments should be repeated and ultrasound and/or calliper measurements used to determine the tumour growth over time.

RESPONSE 9: We accept the reviewer's devil's advocate argument, which is theoretically justified, but not practically. Yes, there are differences in vascularization and perfusion of tumours grown in WT and KO mice. Yet, the BLI measurements have obviously been performed at equilibrium settings. That the BLI measurements are a bona fide readout of tumour is also reflected by the fact that BLI measurements in the 12 day tumours perfectly match the weight difference of harvested tumours at the end of the experiment. The BLI measurements are in fact so robust that we stopped long time ago to validate them with much less accurate calliper measurements.

COMMENT 10: Fig 4: Authors should explain why blood vessels have a small diameter with reduced PC coverage? It would be expected that reduced pericyte association would be associated with an increase in blood vessel diameter.

RESPONSE 10: Indeed, pericyte loss from existing vessels has been reported to be associated with an increase in vessel diameter upon dilatation (14). However, the current setting investigates pericyte function upon active vessel sprouting within a constantly growing tumour. We showed that pericytes are more migratory upon Tie2 deletion and associate less well with endothelial cells. Thus, newly formed vessels could have defects in pericyte association and therefore do not mature into larger diameter vessels explaining the smaller vessel sizes in *Tie2*^{PEKO} animals. The

senior author of the study is a pathologist who has over many years tried to associate parameters of active angiogenesis with morphological vessel patterning (including vessel diameter). There are experimental tumours with grossly deviant vessel morphology. With a grain of salt, it can be said that highly angiogenic tumours (e.g., as assessed by EC proliferation) tend to have smaller diameter microvasculatures. Yet, this is a somewhat subjective statement that we don't want to overinterpret. As such, we have presented the data in the present manuscript as they are and refrained from speculative interpretations.

COMMENT 11: Fig 4 c: It is unclear what 'vessel size (fold)' measurements actually mean and how they were calculated is also unclear. Raw data for at least 40 vessels/ mouse at least 6 mice per genotype should be shown. As it stands the images do not appear to reflect the quantitation on vessel size provided.

RESPONSE 11: The images in **Figure 6d of the revised manuscript** have been replaced by more representative images). Concerning the quantitation, "vessel size (fold)" means average vessel size in KO mice normalized to the mean of WT mice. Results were calculated from 20x sections from 3 different layers each as a tumour tile scan showing the complete tumour area (see Figure B below for an example). This has been clarified in the methods section of the revised manuscript.

Figure B: Tile scan of NG2 coverage in tumours of in *Ng2-Tie2* mice. Representative low magnification overview images of tumours stained for CD31 and NG2 in WT (left) and *Tie2*^{PEKO} (right) mice. Scale bar = 100 μ m.

COMMENT 12: Fig 4 f-h: The authors suggest that the increase in blood vessel density is sufficient to enhance the tumour growth in the KO mice. However, given the reduction in vessel diameter what evidence do the authors have that the vessels are functional and not just leaky. The microsphere data images are very poor and it is almost impossible to count vessels that contain spheres. These should be replaced with better quality images. Indeed, these data rather show leakage of spheres into the tumour mass and do not reflect actual functionality of the vessels. Functional perfusion studies using tail vein administered PE-PECAM-ab would determine whether the KO vessels, despite their smaller size, are well perfused or not. What is the value of the data in Fig h. A 1% difference seems hardly biologically significant.

RESPONSE 12: We have inserted arrows in the microsphere images to better highlight microspheres within and outside of tumour vessels (**Fig. 6f of the revised manuscript**). We agree with the reviewer that the small diameter microsphere experiments (100 nm) first and foremost reflect permeability. This has been clarified in the revised manuscript. Yet, permeability and perfusion are dependent on each other insofar as permeability can only occur from perfused vessels. We have extended our analyses by transmission electron microscopic experiments showing erythrocytes as sign of perfusion in larger and smaller diameter microvessels (**Supplementary Fig. 11 a,b of revised the manuscript**). Lastly, we apologize for the erroneous labelling in Figure 6 h. This is, of course, not a 1% difference, but a doubling of the sphere area compared to total area in the KO mice compared to the WT mice. This has been explicitly stated in the figure legend of the revised manuscript.

COMMENT 13: Fig 4e – Again how was this measured. Is this the number of NG2-positive cells/endothelial cell, as the axis label suggests, or the number of pericytes associated with individual tumour blood vessels? If it is the latter the data should be given as raw data, not fold change. There is one outlier in data – if removed does it affect the statistics? Is this data point skewing the data to make a significance that may not be there if more samples were added to the experiment.

RESPONSE 13: See also response to comment 6: Following the reviewer's advice, we have re-analysed all data with more appropriate non-parametric statistics. Indeed, reflecting the outlier status of one value, the maturation data shown in Fig. 6e are just below significance. This has been corrected in the revised manuscript.

COMMENT 14: Fig 5: Evidence for equal numbers of pericytes in these assays needs to be confirmed histologically. The data should be supported by aortic ring data from the WT and KO mice. Why did the authors not simply carryout *in vivo* matrigel plug assays? Which growth factors are involved here? Growth factor induced angiogenesis assays would be much more informative and would help to develop a more robust mechanism.

RESPONSE 14: Equal numbers of pericytes and endothelial cells were mixed in the co-culture experiments making histological tracing of cell numbers dispensable (which would technically also be somewhat challenging given the 3D nature of the spheroids). As for the reviewer's suggestion to employ other assays such as a Matrigel plug or an aortic ring assay, we would – with all due respect – kindly insist that the *in vitro* and *in vivo* single cell and co-culture assays developed by our laboratory (3,15,16) are way superior to the suggested other assays. The aortic ring assay does not reflect sequential endothelial and pericyte outgrowth. Instead, SMC (and even fibroblasts) outrun in this assay the outgrowing endothelial cells. It would not be clear what would be learnt from this assay in terms of EC-pericyte crosstalk. Likewise, the Matrigel plug assay is a very poor readout of angiogenesis. It yields some findings, but, generally speaking, it is a somewhat unrealistic approach to soak a Matrigel plug with a cytokine and assume that this would yield a sustained effect in a 10-day experiment. Any cytokine will rapidly diffuse away without the use of a slow release pellet. In fact, a 'clean' Matrigel assay under non-inflammatory conditions will hardly result in ingrowth of blood vessels. The senior author of the current manuscript is aware of hundreds of papers published with the Matrigel assay. Yet, based on 20-year hands on experience with this assay, he knows that results obtained in this assay need to be considered with a grain of salt. In contrast, the *in vitro* and *in vivo* spheroid assays employed in the current manuscript are extremely controlled and yield very definite findings. In fact, we believe that the experiment shown in Figure 3c and 3d most elegantly demonstrate the angiogenesis-restraining effect of Tie2 expressing pericytes compared to Tie2 silenced pericytes – which is the only variable in this experiment.

COMMENT 15: Fig 6 –The cell confluency looks very different in shCtr versus shTIE2 I and II, could this reflect the wound closure results?

RESPONSE 15: Cell numbers in the two experimental conditions are identical. We have replaced the images by higher contrast images (Figure 2a of revised manuscript).

COMMENT 16: Densitometric analysis of Fig. 6d is required for repeat experiments.

RESPONSE 16: Figure 6d from the original manuscript has been removed from the revised manuscript to instead include data focussing on the effect of pericyte-specific Tie2 deletion on pericyte-endothelial interactions and intercellular signalling. A detailed description of these data is given below in response to comment 17.

COMMENT 17: Fig 6 e-h – it is really unclear how this potential molecular mechanism hangs together. Overall the mechanism of reduced migration of Tie2kd pericytes is not clear and how this relates to the *in vivo* phenotype is not addressed accept hypothetically. A substantial improvement in the molecular mechanism is required.

RESPONSE 17: To gain further insights into the molecular mechanisms of Tie2 signalling in pericytes and on EC-pericyte intercellular signalling, we performed phospho-receptor tyrosine kinase array experiments of BP and HUVEC co-cultures. Eph receptor and ephrin ligand signalling has been implicated in vascular development and in *in vivo* models of angiogenesis (17). Bi-directional signalling mediates juxtacrine cell-cell contact, cell adhesion to extracellular matrix and cell migration (18-23). Furthermore, signalling in mural cells has been shown to control cell

motility and adhesion as well as pericyte-EC assembly (24,25). Both, EC and pericytes expressed Eph receptors and ephrin ligands (**Supplementary Figure 5 of the revised manuscript**). Co-culture of Tie2-silenced pericytes with EC resulted in reduced EphA2, EphB2 and EphB4 phosphorylation in the co-culture lysates (**Figure 2h of revised manuscript**). Similarly, EphA2, EphB2 and EphB4 phosphorylation was reduced in whole tumour lysates of *Tie2^{PEKO}* animals (**Supplementary Figure 10 of the revised manuscript**). Altogether, deletion of pericyte-expressed Tie2 is accompanied by pro-migratory behaviour and decreased Eph signalling leading to immature vessels with reduced pericyte coverage. Future work will focus on the detailed mechanisms of Tie2 signalling in pericytes. For the purpose of the present study, we believe that these additional data show quite unambiguously that Tie2 is a signalling receptor in pericytes controlling functionally relevant pathways that are compatible with the observed cellular and *in vivo* phenotypes.

COMMENT 18: Fig. 6g – nuclear and cytoplasmatic fractions of FOXO3A with appropriate controls would be more informative.

RESPONSE 18: Phosphorylation of FOXO3A by protein kinases causes its sequestration into the cytoplasm preventing the transcription of target genes within the nucleus. Relocalization to the nuclear protein is achieved via dephosphorylation by protein phosphatases (6). Thus, Western blot analysis of phospho-FOXO3A compared to total FOXO3A can be used to show increased or reduced levels of active FOXO3A. Furthermore, we also show the relocalization of FOXO3A into the nucleus that has been shown to go in line with the active state of FOXO3A (Figure 2g of revised manuscript).

COMMENT 19: Since the authors identify a role for pericyte Tie2 in the control of regulators of invasion have the authors tried metastasis experiments; esp. as they see an increase in microsphere in tumour in KO mice.

RESPONSE 19: Indeed, metastasis analysis has been performed and is now included in the manuscript (**Supplementary Fig. 11c,d in the revised manuscript**). Tie2 knockout mice demonstrated more animals with metastasis formation in the lung. However, due to the primary tumour growth phenotype it is not possible to exclude that this effect rather reflects differences in tumour size than changes in the tumour vasculature.

COMMENT 20: Authors should explain more the relevance of including VECADTie2 embryo data? This seems to be an aside and nothing to do with the pericyte MS and perhaps not that novel.

RESPONSE 20: We establish in the present manuscript that pericyte expressed Tie2 is functional and that it plays a role in developmental and in pathological angiogenesis. As such, the *VE-Cad-Cre* mediated KO of Tie2 is not part of the story. Yet, this is an important control and validation experiment to establish that the embryonic lethal phenotype of globally Tie2 KO mice, as reported more than 20 years ago (26,27), is indeed due to endothelial Tie2 expression and not resulting from a combination of endothelial, pericyte and macrophage expression. We consider it therefore appropriate and necessary to include these control and validation data of the floxed Tie2 mice into the present manuscript.

COMMENT 21: The size of the FAK band in Fig 6d is incorrect. The bands highlighted are less than 100kDa based on the ladder (sup fig 9 for fig 6d). FAK should be approximately 125kDa. The bands on the gel at around 130kd are more likely to be the FAK band and here FAK appears to go up in the pAng2 lane and down in the shAng2 lane thus changing the interpretation of the molecular mechanism of how Tie2 controls pericyte behavior. The authors are advised to revisit these data as they don't seem robust as they stand.

RESPONSE 21: See also response to comment 16. The previous Figure 6d has been removed from the revised manuscript to include data focussing on the effect of pericyte-specific Tie2 deletion on pericyte-endothelial interactions and intercellular signalling. A detailed description of these data is given in response to comment 17.

References

1. Abdel-Malak NA, *et al.*: Angiopoietin-1 promotes endothelial cell proliferation and migration through AP-1-dependent autocrine production of interleukin-8. *Blood* 111, 4145-54 (2008).
2. Schubert SY, Benarroch A, Monter-Solans J, Edelman ER: Primary monocytes regulate endothelial cell survival through secretion of angiopoietin-1 and activation of endothelial Tie2. *Arterioscler Thromb Vasc Biol* 31, 870-5 (2011).
3. Alajati, A. *et al.*: Spheroid-based engineering of a human vasculature in mice. *Nat Methods* 5, 439-45 (2008).
4. St Croix B, *et al.*: Genes expressed in human tumor endothelium. *Science* 289, 1197-202 (2000).
5. Christian S, *et al.*: Endosialin (Tem1) is a marker of tumor-associated myofibroblasts and tumor vessel-associated mural cells. *Am J Pathol* 172, 486-94 (2008).
6. Calnan DR, Brunet A: The FoxO code. *Oncogene* 27, 2276-88 (2008).
7. Crisan, M. *et al.*: A perivascular origin for mesenchymal stem cells in multiple human organs. *Cell stem cell* 3, 301-13 (2008).
8. Crisan, M. *et al.*: Purification and culture of human blood vessel-associated progenitor cells. *Current protocols in stem cell biology* Chapter 2, Unit 2B 2 1-2B 2 13 (2008).
9. Crisan, M. *et al.*: Purification and long-term culture of multipotent progenitor cells affiliated with the walls of human blood vessels: Myoendothelial cells and pericytes. *Methods in Cell Biology* 86, 295-309 (2008).
10. Armulik A, Genove G, Betsholtz C: Pericytes: Developmental, physiological, and pathological perspectives, problems, and promises. *Dev Cell* 21, 193-215 (2011).
11. Göritz C, Dias DO, Tomilin N, Barbacid M, Shupliakov O, Frisén J: A pericyte origin of spinal cord scar tissue. *Science* 333, 238-42 (2011).
12. Saharinen P, *et al.*: Angiopoietins assemble distinct Tie2 signalling complexes in endothelial cell-cell and cell-matrix contacts. *Nat Cell Biol* 10, 527-37 (2008).
13. Boroujerdi A, Tigges U, Welsch-Alves JV, Milner R: Isolation and culture of primary pericytes from mouse brain. *Methods Mol Biol* 1135, 383-92 (2014).
14. Bergers G, Song S: The role of pericytes in blood-vessel formation and maintenance. *Neurooncology* 7, 452-64 (2005).
15. Laib AM, *et al.*: Spheroid-based human endothelial cell microvessel formation in vivo. *Nat Protoc* 4, 1202-15 (2009).
16. Korff T, Augustin HG: Tensional forces in fibrillar extracellular matrices control directional capillary sprouting. *J Cell Sci* 112, 3249-58 (1999).
17. Cheng N, Brantley DM, Chen J: The ephrins and Eph receptors in angiogenesis. *Cytokine Growth Factor Rev* 13, 75-85 (2002).
18. Adams RH, Alitalo K: Molecular regulation of angiogenesis and lymphangiogenesis. *Nat Rev Mol Cell Biol* 8, 464-78 (2007).
19. Adams RH, *et al.*: The cytoplasmic domain of the ligand ephrinB2 is required for vascular morphogenesis but not cranial neural crest migration. *Cell* 104, 57-69 (2001).
20. Bochenek ML, Dickinson S, Astin JW, Adams RH, Nobes CD: Ephrin-B2 regulates endothelial cell morphology and motility independently of Eph-receptor binding. *J Cell Sci* 123, 1235-46 (2010).
21. Fuller T, Korff T, Kilian A, Dandekar G, Augustin HG: Forward EphB4 signaling in endothelial cells controls cellular repulsion and segregation from ephrinB2 positive cells. *J Cell Sci* 116, 2461-70 (2003).
22. Klein R: Eph/ephrin signaling in morphogenesis, neural development and plasticity. *Curr Opin Cell Biol* 16, 580-9 (2004).
23. Wang Y, *et al.*: Ephrin-B2 controls VEGF-induced angiogenesis and lymphangiogenesis. *Nature* 465, 483-6 (2010).
24. Foo SS, *et al.*: Ephrin-B2 controls cell motility and adhesion during blood-vessel-wall assembly. *Cell* 124, 161-73 (2006).
25. Salvucci O, *et al.*: EphrinB reverse signaling contributes to endothelial and mural cell assembly into vascular structures. *Blood* 114, 1707-16 (2009).
26. Dumont DJ, *et al.*: Dominant-negative and targeted null mutations in the endothelial receptor tyrosine kinase, tek, reveal a critical role in vasculogenesis of the embryo. *Genes Dev* 8, 1897-1909 (1994).
27. Sato TN, *et al.*: Distinct roles of the receptor tyrosine kinases Tie-1 and Tie-2 in blood vessel formation. *Nature* 376, 70-74 (1995).

REVIEWERS' COMMENTS:

Reviewer #1 (Remarks to the Author):

The authors have adequately addressed my concerns from the original review.

Reviewer #2 (Remarks to the Author):

None

Reviewer #3 (Remarks to the Author):

I would like to congratulate the authors on the addition of a substantial amount of clarification and new data. This reviewer is satisfied that the paper is now a strong body of work that deserves to be published.